# DiffTORI: Differentiable Trajectory Optimization for Deep Reinforcement and Imitation Learning

**Weikang Wan**[1]*,    **Ziyu Wang**[2]*,    **Yufei Wang**[3]*,
**Zackory Erickson**[3],    **David Held**[3]

[1] Computer Science and Engineering Department, University of California San Diego
[2] Institute for Interdisciplinary Information Sciences, Tsinghua University
[3] Robotics Institute, Carnegie Mellon University
`w2wan@ucsd.edu, ziyu-wan21@mails.tsinghua.edu.cn`
`yufeiw2@andrew.cmu.edu, zackory@cmu.edu, dheld@andrew.cmu.edu`

## Abstract

This paper introduces DiffTORI, which utilizes **Dif**ferentiable **T**rajectory **O**ptimization as the policy representation to generate actions for deep **R**einforcement and **I**mitation learning. Trajectory optimization is a powerful and widely used algorithm in control, parameterized by a cost and a dynamics function. The key to our approach is to leverage the recent progress in differentiable trajectory optimization, which enables computing the gradients of the loss with respect to the parameters of trajectory optimization. As a result, the cost and dynamics functions of trajectory optimization can be learned end-to-end. DiffTORI addresses the "objective mismatch" issue of prior model-based RL algorithms, as the dynamics model in DiffTORI is learned to directly maximize task performance by differentiating the policy gradient loss through the trajectory optimization process. We further benchmark DiffTORI for imitation learning on standard robotic manipulation task suites with high-dimensional sensory observations and compare our method to feed-forward policy classes as well as Energy-Based Models (EBM) and Diffusion. Across 15 model-based RL tasks and 35 imitation learning tasks with high-dimensional image and point cloud inputs, DiffTORI outperforms prior state-of-the-art methods in both domains.

## 1   Introduction

Recent works have shown that the representation of a policy can have a substantial impact on the learning performance [1; 2; 3; 4]. Prior works have explored the use of feed-forward neural networks [4], energy-based models [2], or diffusion [1; 5] as the policy representation. In this paper, we propose to use differentiable trajectory optimization [3; 6; 7; 8; 9] as the policy representation to generate actions for deep reinforcement learning (RL) and imitation learning (IL) with high-dimensional sensory observations (images/point clouds).

Trajectory optimization is an effective and widely used algorithm in control, defined with a cost function and a dynamics function. It can be viewed as a policy [3; 6], where the parameters of the policy specify the cost function and the dynamics function. Given the learned cost and dynamics functions as well as the input state (e.g., images, point clouds, robot joint states), the policy then computes the actions by solving the trajectory optimization problem. Trajectory optimization can also be made to be differentiable, which allows back-propagating through the trajectory optimization process [3; 8; 10; 6; 9; 11; 12; 13]. In prior work, differentiable trajectory optimization has been

---

*Equal contribution. This work was performed when Weikang Wan and Ziyu Wang were interning at CMU.

38th Conference on Neural Information Processing Systems (NeurIPS 2024).

applied to system identification [3; 6; 9], inverse optimal control [6], imitation learning [3; 6; 8; 14; 7] and control/planning for robotics problems with low-dimensional states [3; 6; 8; 15].

In this paper, we propose to combine differentiable trajectory optimization with deep model-based RL algorithms. Because we use differentiable trajectory optimization to generate actions [10], we are able to compute the policy gradient loss on the generated actions to learn the dynamics and cost functions to optimize the reward. This approach addresses the "objective mismatch" issue [16; 17] of current model-based RL algorithms, i.e. models that achieve better training performance (e.g., lower MSE) in learning a dynamics model are not necessarily better for control. Our method addresses this issue, as the latent dynamics and reward models are both optimized to maximize the task performance by back-propagating the policy gradient loss through the trajectory optimization process. We show that our method outperforms prior state-of-the-art model-based RL algorithms on 15 tasks from the DeepMind Control Suite [18] with high-dimensional image inputs.

We further benchmark our method for imitation learning on standard robotic manipulation task suites with high-dimensional sensory observations and compare our method to feed-forward policy classes as well as Energy-Based Models (EBM) [2] and Diffusion [1], and term our method DiffTORI (**Diff**erentiable **T**rajectory **O**ptimization for **R**einforcement and **I**mitation Learning). We observe that our training procedure using differentiable trajectory optimization leads to better performance compared to the EBM approach used in prior work, which can suffer from training instability due to the requirement of sampling high-quality negative examples [1]. We also outperform diffusion-based approaches [1] due to our procedure of learning a cost function that we optimize at test time. We show DiffTORI achieves state-of-the-art performance across 35 different tasks: 5 tasks from Robomimic [19] with image inputs, 9 tasks from Maniskill1 [20] and Maniskill2 [21] with point cloud inputs, and 22 tasks from MetaWorld [22] with point cloud inputs.

Our work is closely related to prior work [3; 8; 6] in employing differentiable trajectory optimization as a policy representation. Compared to these prior work, we are the first to show how differentiable trajectory optimization can be combined with deep model based RL algorithms, training dynamics, reward, Q function, and the policy end-to-end using task loss. In contrast, prior work either focuses on imitation learning [3; 8], assumes known dynamics and reward structures and learns only a few parameters [3], or first learns the dynamics model with the dynamics prediction loss (instead of the task loss), and then uses the fixed learned dynamics for control [8]. We are also the first to show that the policy class represented by differentiable trajectory optimization can scale up to high-dimensional sensory observations like images and point clouds, achieving state-of-the-art performances in standard RL and imitation learning benchmarks. In contrast, prior works [3; 8; 6] only test their methods in customized tasks with ground-truth low-level states, and do not report performance on standard benchmarks with more complex tasks and high-dimensional observations.

In summary, the contributions of our paper are as following:

- We introduce DiffTORI, which uses differentiable trajectory optimization as the policy representation for deep reinforcement learning and imitation learning.
- We conduct extensive experiments to compare DiffTORI against prior state-of-the-art methods on 15 tasks for model-based RL and 35 tasks for imitation learning in standard benchmarks with high-dimensional sensory observations, and show that DiffTORI achieves superior performances.
- We perform analysis and ablations of DiffTORI to provide insights into its performance gains.

## 2 Related Works

**Differentiable optimization and implicit policy representation:**  Our work follows the line of research on differentiable optimization, which embeds optimization problems as a layer in neural networks for end-to-end learning. Early works focus on differentiating through convex optimization problems [23; 24]. Recent works extend the range of optimization problems that can be made differentiable [11; 12; 6; 8; 9; 10]. The mostly related prior work to ours are Amos et al. [3] and Jin et al. [6], which first proposed to treat trajectory optimization as an implicit policy and demonstrated its effectiveness in the setting of behavior cloning, system identification, and control for robotics problems with low-dimensional states. Another closely related recent work is Romero et al. [15], where they embed a differentiable quadratic program with learnable cost matrices and known dynamics into the last layer of the actor in PPO, with applications for quadcopter flying. Ours differ

from this work as we learn non-linear costs parameterized by a full neural network, and we also learn the dynamics instead of assuming it is known. We also show our method work with high-dimensional sensory inputs such as images and point clouds. Cheng et al. [25; 26] proposes to learn the parameters of a PID controller by unrolling the controller and system dynamics into a computation graph and optimizing the controller parameters via gradient descent with respect to the task loss, assuming known dynamics. DiffTORI does not assume any prior knowledge on the dynamics or policy class; Instead of representing the policy as a predefined controller, our policy is represented as performing trajectory optimization with the learned dynamics, reward and Q functions represented as neural networks. Sacks et al. [27] proposes to learn the update rule in MPPI, represented as a neural network, using reinforcement learning, with known dynamics and cost functions. Instead of learning the update rule, we learn the dynamics, reward, Q function used in trajectory optimization to generate the actions. We perform differentiable trajectory optimization instead of RL to optimize the parameters of these functions. Differentiable optimization has also been applied in other robotics domains such as autonomous driving [14; 28; 29], navigation [7; 30], motion planning [31; 12], and state estimation [32]. We are the first to show how differentiable trajectory optimization can be combined with deep model-based RL.

**Model-based reinforcement learning:** Compared to model-free RL, model-based RL usually has higher sample efficiency as it is solving a simpler supervised learning problem when learning the dynamics model. Recently, researchers have identified a fundamental problem for model-based RL, known as "objective mismatch" [16]. Recent works have proposed a single objective which is a lower bound on the true return of the policy, for joint model and policy learning in model-based RL [17; 33]. Our approach also addresses the objective mismatch problem. In contrast to these two prior work which only optimizes a lower bound on the true return, our approach directly optimizes the task reward. Further, these approaches are only demonstrated using low-dimensional state-based observations whereas our approach is able to handle high-dimensional image or point cloud observations. In contrast to these works, we use Theseus [10] to analytically compute the gradient of the true objective for updating the model. Another related work, Nikishin et al. [34] proposes to learn a dynamics and reward model in model-based RL, and derive an implicit policy as the softmax policy associated with the optimal Q function under the learned dynamics and reward, learned by back-propagating the RL loss via implicit function theorem. In contrast, we derive the implicit policy as the optimal solution from performing trajectory optimization with the learned dynamics, reward and Q function.

**Policy architecture for deep imitation learning:** Imitation learning can be formulated as the supervised regression task of learning to map observations to actions from demonstrations. Some recent work explores different policy architectures (e.g., explicit policy, implicit policy [2], diffusion policy [1]) and different action representations (e.g., mixtures of Gaussian [35; 19], spatial action maps [36], action flow [4], or parameterized action spaces [37]) to achieve more accurate learning from demonstrations, to model the multimodal distributions of demonstrations, and to capture sequential correlation. Our method outperforms explicit or diffusion policy approaches due to our procedure of learning a cost function that we optimize at test time. In comparison with the implicit policy, which also employs test-time optimization with a learned obective, we use a different and more stable training procedure via differentiable trajectory optimization.

## 3 Background

### 3.1 Differentiable Trajectory Optimization

In robotics and control, trajectory optimization solves the following type of problems:

$$\min_{a_0,\dots,a_T} \sum_{t=0}^{T-1} c(s_t, a_t) + C(s_T) \tag{1}$$
$$s.t. \quad s_{t+1} = d(s_t, a_t)$$

where $c(s_t, a_t)$ and $C(s_T)$ are the cost functions, and $s_{t+1} = d(s_t, a_t)$ is the dynamics function. In this paper, we consider the case where the cost function and the dynamics functions are neural networks parameterized by $\theta$: $c_\theta(s_t, a_t)$, $C_\theta(s_T)$, and $d_\theta(s_t, a_t)$.

Let $a_0(\theta), ..., a_T(\theta)$ be the optimal solution to the trajectory optimization problem, which is a function of the model parameters $\theta$. Differentiable trajectory optimization is a class of method that enables

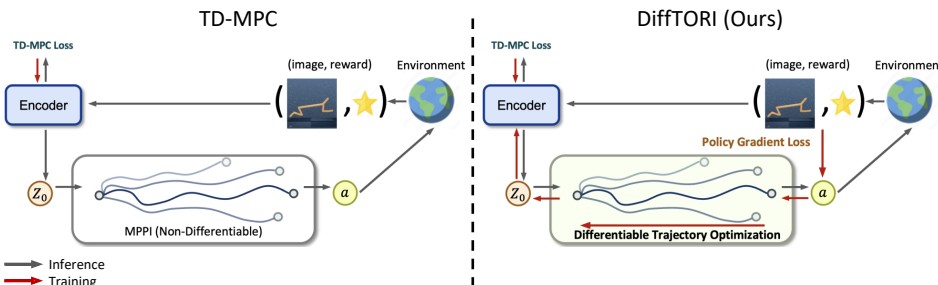

Figure 1: **Overview of DiffTORI for model-based RL**. In contrast to prior work in model-based RL [38] that uses non-differentiable MPPI (left), we utilize differentiable trajectory optimization to generate actions (right). DiffTORI computes the policy gradient loss on the generated actions and back-propagates it through the optimization process, to optimize the encoder as well as other latent space models (latent reward predictor and latent dynamics function) to maximize task performance.

computation of the gradient of the actions with respect to the model parameters $\frac{\partial a_t(\theta)}{\partial \theta}$. Specifically, in this paper we use Theseus [10], which is an efficient application-agnostic open source library for differentiable nonlinear least squares optimization. Theseus works well with high-dimensional states, e.g., images or point clouds, along with using neural networks as the cost and dynamics functions.

## 3.2 Model-Based RL preliminaries

We use the standard MDP formulation: $\langle \mathcal{S}, \mathcal{A}, \mathcal{R}, \mathcal{T}, \gamma \rangle$ where $\mathcal{S}$ is the state space, $\mathcal{A}$ is the action space, $\mathcal{R}(s, a)$ is the reward function, $\mathcal{T}(\cdot|s, a)$ is the transition dynamics function, and $\gamma \in [0, 1)$ is the is the discount factor. The goal is to learn a policy $\pi$ to maximize the expected return: $\mathbb{E}_{s_t, a_t \sim \pi}[\sum_{t=1}^{\infty} \gamma^t R(s_t, a_t)]$. In this paper we work on problems where the state space $S$ are high-dimensional sensory observations, e.g., images or point clouds. Model-based RL algorithms first learn a dynamics model, and then use it for learning a policy. When applied to model-based RL, our method builds upon TD-MPC [38], a recently proposed model-based RL algorithm which we review briefly here. We choose TD-MPC for its simplicity and state-of-the-art performance. However, our method is compatible with any model-based RL algorithm that learns a dynamics model and a reward function. TD-MPC consists of the following components: first, an encoder $h_\theta$, which encodes the high-dimensional sensory observations, e.g., images, into a low-dimensional state $z_t = h_\theta(s_t)$. In the latent space, a latent dynamics model $d_\theta$ is also learned: $z_{t+1} = d_\theta(z_t, a_t)$. A latent reward predictor $R_\theta$ is learned which predicts the task reward $r$: $\hat{r} = R_\theta(z_t, a_t)$. Finally, a value predictor $Q_\theta$ learns to predict the Q value: $\hat{Q} = Q_\theta(z_t, a_t)$. Note that we use $\theta$ to denote all learnable parameters including the encoder, the latent dynamics model, the reward predictor, and the Q value predictor. These models are trained jointly using the following objective:

$$\mathcal{L}_{\text{TD-MPC}}(\theta; \tau) = \sum_{i=t}^{t+H} \lambda^{i-t} \mathcal{L}_{\text{TD-MPC}}(\theta; \tau_i),\qquad(2)$$

where $\tau \sim \mathcal{B}$ is a trajectory $(s_t, a_t, r_t, s_{t+1})_{t:t+H}$ sampled from a replay buffer $\mathcal{B}$, $\lambda \in \mathbb{R}_+$ is a constant that weights near-term predictions higher, and the single-step loss is:

$$\mathcal{L}_{\text{TD-MPC}}(\theta; \tau_i) = c_1 \underbrace{\|R_\theta(\mathbf{z}_i, \mathbf{a}_i) - r_i\|_2^2}_{\text{reward}} + c_2 \underbrace{\|Q_\theta(\mathbf{z}_i, \mathbf{a}_i) - (r_i + \gamma Q_{\theta-}(\mathbf{z}_{i+1}, \pi_\theta(\mathbf{z}_{i+1})))\|_2^2}_{\text{value}}$$
$$+ c_3 \underbrace{\|d_\theta(\mathbf{z}_i, \mathbf{a}_i) - h_{\theta-}(\mathbf{s}_{i+1})\|_2^2}_{\text{latent state consistency}}\qquad(3)$$

where $\theta^-$ are parameters of target networks that are periodically updated using the parameters of the learning networks. As shown in (3), the parameters $\theta$ is optimized with a set of surrogate losses (reward prediction, value prediction, and latent consistency), rather than directly optimizing the task performance, known as the objective mismatch issue [16]. At test time, model predictive path integral (MPPI) [39] is used for planning actions that maximize the predicted rewards and Q functions in the latent space. A policy $\pi_\psi$ is further learned in the latent space using the latent Q-value function, which is used to generate action samples in the MPPI process.

# 4 Method

## 4.1 Overview

The core idea of DiffTORI is to use trajectory optimization as the policy $\pi_\theta$, where $\theta$ parameterizes the dynamics and cost functions. Given a state $s$, DiffTORI generates the actions $a(\theta)$ by solving the trajectory optimization problem in (1) with $s_0 = s$. To optimize the policy parameters $\theta$, we use differentiable trajectory optimization to compute the gradients of the loss $\mathcal{L}(a(\theta))$ with respect to the policy parameters: $\frac{\partial \mathcal{L}(a(\theta))}{\partial \theta}$, where the exact form of the loss depends on the problem setting.

An overview of applying DiffTORI to model-based RL is shown in Figure 1. Existing model-based RL algorithms such as TD-MPC suffer from the objective mismatch issue: the latent dynamics and reward (cost) functions are learned to optimize a set of surrogate losses (as in (3)), instead of optimizing the task performance directly. DiffTORI addresses this issue: by computing the policy gradient loss on the optimized actions from trajectory optimization and differentiating through the trajectory optimization process, the dynamics and cost functions are optimized directly to maximize the task performance. We describe DiffTORI for model-based RL in Section 4.2.

We also apply DiffTORI to imitation learning; an overview is shown in Figure 2. In contrast to explicit policies that generate actions at test-time by forward passes of the policy network, DiffTORI generates the actions via test-time trajectory optimization with a learned cost function. This is in the same spirit of implicit behaviour cloning [2] which learns an energy function and optimizes with respect to it to generate actions at test-time. However, we observe that our training procedure using differentiable trajectory optimization leads to better performance compared to the EBM approach used in prior work, which can suffer from training instability due to the requirement of sampling high-quality negative examples [1]. We describe DiffTORI for imitation learning in detail in Section 4.3.

## 4.2 Differentiable trajectory optimization applied to model-based RL

We build DiffTORI on top of TD-MPC for model-based RL. Similar to TD-MPC, DiffTORI consists of an encoder $h_\theta$, a latent dynamics model $d_\theta$, a reward predictor $R_\theta$, and a Q-value predictor $Q_\theta$ (see Sec. 3.2). We use $\theta$ to denote all learnable parameters to be optimized in DiffTORI. As shown in Figure 1, the key to DiffTORI is to change the non-differentiable MPPI planning algorithm in TD-MPC to a differentiable trajectory optimization, and include the policy gradient loss on the generated actions to optimize the model parameters $\theta$ directly for task performance.

Formally, given a state $s_t$, we use the encoder $h_\theta$ to encode it to the latent state $z_t$, and then construct the following trajectory optimization problem in the latent space:

$$a(\theta) = \underset{a_t,\ldots,a_{t+H}}{\arg\max} \sum_{l=t}^{H-1} \gamma^{l-t} R_\theta(z_t, a_t) + \gamma^H Q_\theta(z_H, a_H) \tag{4}$$

$$s.t. \ z_{t+1} = d_\theta(z_t, a_t)$$

where $H$ is the planning horizon. In this paper we leverage Theseus [10] to solve (4) in a differentiable way. Since Theseus only supports solving non-linear least-square optimization problems without constraints, we remove the dynamics constraints in the above optimization problem by manually rolling out the dynamics into the objective function. For example, with a planning horizon of $H = 2$, we turn the above optimization problem into the following one:

$$a(\theta) = \underset{a_t, a_{t+1}, a_{t+2}}{\arg\max} \ R_\theta(z_t, a_t) + R_\theta(d_\theta(z_t, a_t), a_{t+1}) + Q_\theta(d_\theta(d_\theta(z_t, a_t), a_{t+1}), a_{t+2}) \tag{5}$$

We set the values of $H$ following the schedule as in TD-MPC, and we use the Levenberg–Marquardt algorithm in Theseus to solve the optimization problem. Following TD-MPC, we also learn a policy $\pi_\psi$ in the latent space using the learned Q-value predictor $Q_\theta$, and the output from the policy is used as the action initialization for solving (4).

Let $a(\theta)$ be the solution of the above trajectory optimization problem, obtained using Theseus as described above. DiffTORI is learned with the following objective, which jointly optimizes the encoder, latent dynamics model, latent reward model, and the Q-value predictor:

$$\mathcal{L}_{DiffTORI}^{RL}(\theta; \tau) = \sum_{i=t}^{t+H} \lambda^{i-t} \left( \mathcal{L}_{TD-MPC}(\theta; \tau_i) + c_0 \mathcal{L}_{PG}(\theta; \tau_i) \right) \tag{6}$$

$$\mathcal{L}_{PG}(\theta; \tau_i) = -\tilde{Q}_\phi(s_i, a(\theta))$$

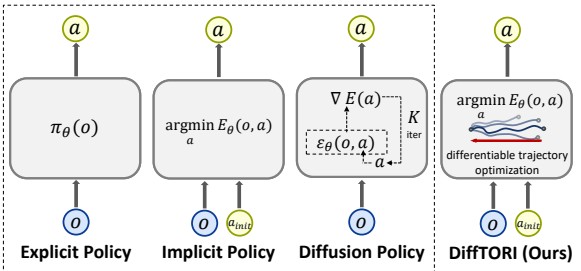

Figure 2: **Overview of our method on Imitation Learning.** DiffTORI (right) learns a cost function via differentiable trajectory optimization and performs test-time optimization with it, which is different from prior work (left) that uses an explicit policy or diffusion without test-time optimization. Although implicit policy shares the same spirit as DiffTORI, we observe that the training procedure of DiffTORI using differentiable trajectory optimization leads to better performance compared to the EBM approach used in prior work [2], which can suffer from training instability.

where $\tilde{Q}_\phi$ is the Q function learned via Bellman updates [40] which is used to compute the deterministic policy gradient [41], and $c_0$ is the weight for this loss term. $\tilde{Q}_\phi$ is learned in the original state space $\mathcal{S}$ instead of the latent space to provide accurate policy gradients. The key idea here is that we can backpropagate through the policy gradient loss $\mathcal{L}_{PG}$, which backpropagates through $a(\theta)$ and then through the differentiable trajectory optimization procedure of Equation 4 to update $\theta$.

### 4.3 Differentiable Trajectory Optimization applied to imitation learning

We also use DiffTORI for model-based imitation learning. A comparison of DiffTORI to other types of policy classes used in prior work is shown in Figure 2. In this approach, DiffTORI consists of an encoder $h_\theta$ and a latent dynamics function $d_\theta$, as before. However, in the setting of imitation learning, we do not assume access to a reward function $\mathcal{R}(s, a)$. Instead, we generate actions by solving the following trajectory optimization problem:

$$a(\theta) = \underset{a_t, \ldots, a_{t+H}}{\arg\max} \sum_{l=t}^{H} \gamma^{l-t} f_\theta(z_t, a_t) \tag{7}$$
$$s.t. \ z_{t+1} = d_\theta(z_t, a_t),$$

in which $f_\theta(z_t, a_t)$ is a function over the latent state $z_t$ and actions $a_t$ that we will optimize using the imitation learning loss, as described below. Similarly, We use $\theta$ to denote all learnable parameters to be optimized in DiffTORI, including the parameters of the encoder $h_\theta$, the latent dynamics model $d_\theta$, and the function $f_\theta$ in the imitation learning setting.

In imitation learning, we assume access to an expert dataset $D = \{(s_i, a_i^*)\}_{i=1}^N$ of state-action pairs $(s_i, a_i^*)$. In the most basic form, the loss $\mathcal{L}$ for DiffTORI can be the mean square error between the the expert actions $a_i^*$ and the actions $a(\theta)$ returned from solving (7):

$$\mathcal{L}_{BC}(\theta) = \sum_{i=1}^N ||a(\theta) - a_i^*|| \tag{8}$$

The key idea here is that we can backpropagate through the imitation loss $\mathcal{L}_{BC}$, which backpropagates through $a(\theta)$ and then through the differentiable trajectory optimization procedure of Equation 7 to update $\theta$. This enables us to learn the function $f_\theta(z_t, a_t)$ used in the optimization Equation 7 directly by optimizing the imitation loss $\mathcal{L}_{BC}(\theta)$. Because this loss is optimized through the trajectory optimization procedure (Equation 7), we will learn a function $f_\theta(z_t, a_t)$ such that optimizing Equation 7 returns actions that match the expert actions.

**Multimodal DiffTORI:** The loss in Equation 8 will not be able to capture multi-modal action distributions in the expert demonstrations. To address this, we use a Conditional Variational Auto-Encoder (CVAE) [42] as the policy architecture, which has the ability to capture a multi-modal action distribution [43]. The CVAE encoder encodes the state $s_i$ and the expert action $a_i^*$ into a latent state vector $z_i$. The key idea in our approach is that the decoder in CVAE takes the form of a trajectory optimization algorithm, given by Equation 7. It takes as input the sampled latent $\tilde{z}$ from the Gaussian Prior, and the state $s_i$ and uses differentiable trajectory optimization to decode the

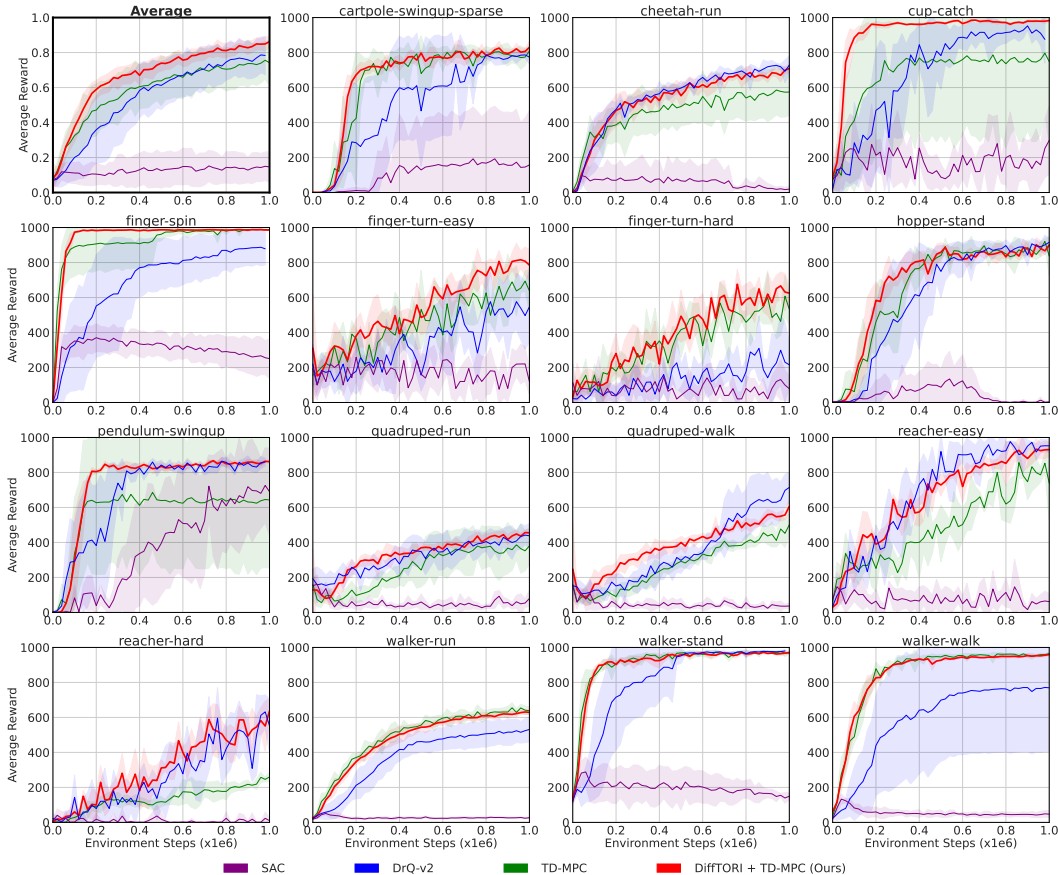

Figure 3: Performance of DiffTORI, in comparison to 4 prior state-of-the-art model-based and model-free RL algorithms, on 15 tasks from DeepMind control suite. DiffTORI achieves the best performance when averaged across all tasks. Results are averaged with 4 seeds, and the shaded regions represent the 95% confidence interval.

action $a(\theta)$. Because this trajectory optimization is differentiable, we can backpropagate through it to learn the parameters $\theta$ for the encoder, dynamics $d_\theta$, and the function $f_\theta$ used in Equation 7. See Appendix D for further details.

**Action refinement:** We note that DiffTORI provides a natural way to perform action refinement on top of a base policy. Given an action from any base policy, we can use this action as the initialization of the action variables for solving the trajectory optimization problem; the trajectory optimizer will iteratively refine this action initialization with respect to the optimization objective of Equation 7. In our experiments, we find DiffTORI always outperforms the base policies when using their actions as the initialization and other ways of performing action refinement, such as residual learning.

## 5 Experiments

### 5.1 Model-based Reinforcement Learning

We conduct experiments on 15 DeepMind Control suite tasks, which involve simulated locomotion and manipulation tasks, such as making a cheetah run or swinging a ball into a cup. All tasks use image observations and the control policy does not have direct access to the underlying states.

We compare to the following baselines: **SAC** [44], a commonly used off-policy model-free RL algorithm. **DrQ-v2** [45], a state-of-the-art model-free RL algorithm for image observations that adds data augmentation on top of SAC . **TD-MPC** [38], a state-of-the-art model-based RL algorithm, which DiffTORI builds on. All training details such as hyper-parameters, and pysudo-code can be found in Appendix B. All experiments use NVIDIA 2080 Ti GPUs.

Figure 3 shows the learning curves for all methods on all tasks. The top-left subplot shows the normalized performance averaged across all 15 tasks, which is computed as the achieved return divided by the max return from any algorithm. As shown, DiffTORI (red curve) outperforms all compared baselines by a noticeable margin. On 14 out of the 15 tasks (except Quadruped-walk), DiffTORI achieves the highest performance among compared algorithms. We especially note that the performance of DiffTORI is much higher than TD-MPC, which DiffTORI builds on, showing the benefit of adding the policy gradient loss and directly differentiating through it to optimize the learned latent spaces. Although DiffTORI achieves higher sample efficiency, one limitation of DiffTORI is that it requires more wall-clock time for training, due to the need for solving and differentiating through the trajectory optimization process. We show detailed results on computational efficiency (return vs wall-clock time) of DiffTORI in Appendix A.1.2. We also perform ablation studies to examine how each loss term in (6) contributes to the final performance of DiffTORI in Figure 6 in Appendix A.1.3.

Table 1: Success rates ($\uparrow$) of DiffTORI, DP3 and Residual + DP3 on 22 MetaWorld tasks. DiffTORI consistently achieves higher or on-par success rates on all 22 tasks.

| | MetaWorld (Medium) | | | | | | |
| --- | --- | --- | --- | --- | --- | --- | --- |
| | Soccer | Push Wall | Peg Insert Side | Bin Picking | Basketball | Box Close | Coffee Pull |
| DP3 | 32.3±3.5 | 42.0±2.6 | 57.3±10.3 | 13.0±1.0 | **88.0**±5.3 | 62.0±2.0 | 65.0±3.6 |
| Residual + DP3 | 33.0±2.6 | 43.3±3.1 | 59.3±4.2 | 13.3±3.1 | 87.3±3.1 | 67.3±5.0 | 59.3±1.5 |
| DiffTORI (Ours) + DP3 | **40.7**±1.2 | **50.0**±2.0 | **64.7**±4.2 | **22.0**±5.3 | **88.0**±4.0 | **73.3**±4.6 | **70.7**±7.0 |

| | Metaworld (Medium) | | | | Metaworld (Hard) | | | |
| --- | --- | --- | --- | --- | --- | --- | --- | --- |
| | Coffee Push | Hammer | Sweep | Sweep Into | Assemble | Hand Insert | Pick out of Hole | Pick Place |
| DP3 | 53.0±3.6 | 32.3±3.5 | 74.7±3.1 | 30.3±11.2 | 68.7±1.5 | 18.3±2.1 | 55.0±4.6 | 56.7±2.5 |
| Residual + DP3 | 49.3±4.2 | 31.3±1.2 | 71.3±3.0 | **45.7**±3.8 | 64.7±3.1 | 19.7±2.1 | 61.3±2.3 | 52.0±2.0 |
| DiffTORI (Ours) + DP3 | **60.7**±5.0 | **37.3**±3.1 | **90.7**±3.1 | 45.3±6.1 | **74.0**±4.0 | **24.7**±1.2 | **63.3**±6.1 | **61.3**±6.3 |

| | Metaworld (Hard) | | Metaworld (Very Hard) | | | | | Average |
| --- | --- | --- | --- | --- | --- | --- | --- | --- |
| | Push | Push Back | Shelf Place | Disassemble | Stick Pull | Stick Push | Pick Place Wall | |
| DP3 | 21.3±7.5 | 55.3±4.9 | 27.7±2.9 | 34.0±4.6 | 53.0±6.9 | **94.3**±2.1 | 38.3±5.7 | 48.8 |
| Residual + DP3 | 20.0±3.5 | 55.3±4.2 | 35.3±4.2 | 35.3±1.2 | 56.0±2.0 | 91.3±2.3 | **44.7**±4.2 | 49.8 |
| DiffTORI (Ours) + DP3 | **30.0**±3.5 | **64.7**±3.1 | **42.0**±8.0 | **40.7**±3.1 | **59.3**±6.1 | **94.0**±3.5 | **44.7**±2.3 | **56.5** |

Table 2: Failure rates ($\downarrow$) of all methods on the Robomimic tasks. DiffTORI achieves the lowest failure rates on all tasks with diffusion policy as the base policy.

| | IBC | BC-RNN | Residual +BC-RNN | DiffTORI (Ours) + BC-RNN | Diffusion | IBC + Diffusion | Residual + Diffusion | DiffTORI (Ours) + Diffusion |
| --- | --- | --- | --- | --- | --- | --- | --- | --- |
| Square | 0.96±0.00 | 0.18±0.00 | 0.16±0.01 | 0.10±0.02 | 0.12±0.03 | 0.32±0.05 | 0.12±0.02 | **0.08**±0.01 |
| Transport | 1.00±0.00 | 0.28±0.03 | 0.26±0.03 | 0.17±0.02 | 0.07±0.04 | 0.92±0.03 | 0.08±0.01 | **0.04**±0.01 |
| ToolHang | 1.00±0.00 | 0.33±0.04 | 0.28±0.03 | 0.18±0.00 | 0.10±0.00 | 0.94±0.01 | 0.10±0.00 | **0.08**±0.01 |
| Push-T | 0.89±0.01 | 0.30±0.02 | 0.28±0.02 | 0.25±0.02 | **0.09**±0.00 | 0.92±0.01 | **0.09**±0.00 | 0.09±0.01 |
| **Average** | 0.96 | 0.27 | 0.25 | 0.18 | 0.10 | 0.78 | 0.10 | **0.07** |

## 5.2 Imitation Learning

Below we show results of DiffTORI on 3 commonly used imitaiton learning benchmarks: Meta-World [22], RoboMimic [19], ManiSkill [20], and the comparison to state-of-the-art methods on these three benchmarks. We also compare to one closely related prior work [3] on one of their customized tasks in Appendix A.3.

### 5.2.1 MetaWorld

MetaWorld [22] is a large-scale benchmark that includes 100 robotic manipulation tasks, and has been recently used for evaluating different imitation learning algorithms [46]. The policy observation is point clouds of the scene, and the action is the 3d translation of the robot end-effector. We test on 22 tasks with different levels of difficulties: Medium, Hard, and Very Hard (See Table 1 for all the tasks). 10 demonstrations are used for all tasks [46]. We compare DiffTORI with the following baselines: **DP3** [46], a 3D version of diffusion policy that achieves state-of-the-art performances on this benchmark, outperforming other algorithms such as the original diffusion policy [1] with 2d image inputs. **Residual + DP3**: Since DiffTORI refines the actions from a base pre-trained DP3 policy, we additionally compare to this baseline that also leverages the actions from a base pre-trained DP3 policy. Specifically, we learn a residual policy on top of the base pre-trained policy, which takes

Table 3: On Maniskill tasks, DiffTORI consistently achieves higher success rates (↑) on all tasks.

| | PickCube | Fill | Hang | Excavate | Pour | OpenCabinet Drawer | OpenCabinet Door | PushChair | MoveBucket | Average |
|---|---|---|---|---|---|---|---|---|---|---|
| BC | 0.19±0.03 | 0.72±0.04 | 0.76±0.02 | 0.25±0.02 | 0.13±0.01 | 0.47±0.03 | 0.35±0.04 | 0.12±0.01 | 0.10±0.01 | 0.34 |
| BC + residual | 0.21±0.04 | 0.75±0.02 | 0.75±0.02 | 0.27±0.03 | 0.12±0.01 | 0.49±0.02 | 0.36±0.03 | 0.15±0.02 | 0.10±0.01 | 0.36 |
| DiffTORI (Ours) + BC | **0.32**±0.02 | **0.82**±0.01 | **0.85**±0.03 | **0.29**±0.01 | **0.17**±0.02 | **0.53**±0.02 | **0.45**±0.02 | **0.20**±0.02 | **0.15**±0.02 | **0.42** |

as input the action from the base policy, and outputs a delta action that is added to the base action. This is the most standard and simple way of doing residual learning. All training details such as hyper-parameters and pseudo-code can be found in Appendix B.

Table 1 presents the task success rates, averaged over 50 evaluation episodes, of all compared algorithms. As shown, DiffTORI consistently achieves higher (or on par) success rates than the other 2 compared baselines. The improvement in success rates is larger on tasks where the original DP3 policy struggles, e.g., a 15% improvement on the task of Shelf Place and Sweep Into; and as expected, when the base DP3 policy is already doing well on the task, there is not much room of improvement left for DiffTORI, e.g., on Basketball and Stick Push. The simple way of learning a residual policy on top of the DP3 policy does not always improve the performance of the base policy, and even leads to lower success rates. This demonstrates that DiffTORI is a more effective way to leverage a pre-trained policy. On average, the success rates of DiffTORI is 7.7% higher than that of DP3, a substantial improvement with only 10 demonstrations.

### 5.2.2 Robomimic

Robomimic [19] is another commonly used benchmark designed to study imitation learning for robot manipulation. The benchmark encompasses a total of 5 tasks with two types of demonstrations: collected from proficient humans (PH) or a mixture of proficient and non-proficient humans. We use the PH demonstrations, and evaluate on three of the most challenging tasks: Square, Transport, and ToolHang. We use image-based observations and the default velocity controller for all the tasks. In addition to Robomimic, we compare to another task, Push-T from the diffusion policy [1] task set, to demonstrate that we can learn multimodal cost functions by using the CVAE training loss.

We compare to the following baselines: **IBC** [2]: An implicit policy that learns an energy function conditioned on both action and observation using the InfoNCE loss [47]. **BC-RNN** [19]: A variant of BC that uses a Recurrent Neural Network (RNN) as the policy network to encode a history of observations. This is the best-performing baseline in the original Robomimic [19] paper. **Residual + BC-RNN**: We use a pretrained BC-RNN as the base policy, and learn a residual policy on top of it. The residual policy takes as input the action from the base policy, and outputs a delta action which is added to the base action. **Diffusion Policy** [1]: A policy that uses the diffusion model as the policy class. It refines noise into actions via a learned gradient field. **IBC + Diffusion**: A version of IBC that uses the action from a pre-trained Diffusion Policy as the action initialization in the test-time optimization process. **Residual + Diffusion**: Similar to Residual + BC-RNN, but using a pre-trained Diffusion Policy as the base policy. For DiffTORI, we compare two variants of it: DiffTORI + BC-RNN and DiffTORI + Diffusion Policy, which uses a pre-trained BC-RNN or a pre-trained diffusion policy as the base policy to generate the initialization action for solving the trajectory optimization problem. In Appendix A.2, we also present results of DiffTORI with zero initialization or random initialization, instead of initializing the action from a base policy.

The results are shown in Table 2. We find that DiffTORI+Diffusion Policy achieves the lowest failure rates consistently across all tasks. Even though Diffusion Policy has almost saturated on these tasks with very low failure rates, DiffTORI can still further reduces it. Furthermore, irrespective of the base policy used — whether BC-RNN or Diffusion Policy — DiffTORI always brings noticeable improvement in the performance over the base policy. While learning a residual policy does lead to improvements upon the base policy, DiffTORI shows a significantly greater performance boost. In addition, by comparing DiffTORI+Diffusion Policy with IBC+Diffusion Policy, we find that using the same action initialization for IBC is considerably less effective than using the same action initialization in DiffTORI. In many tasks, even when the base Diffusion Policy already exhibits low failure rates, IBC+Diffusion Policy still results in poor performances, indicating the training objective used in IBC actually deteriorates the base actions.

We also show the benefit of using a CVAE architecture for DiffTORI, which enables DiffTORI to capture multimodal action distributions. With different latent samples from CVAE, we get

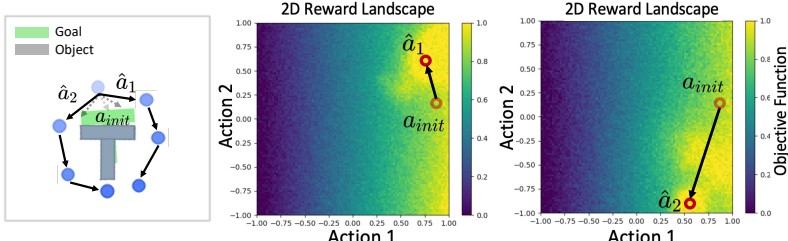

Figure 4: By using a CVAE, DiffTORI can learn multimodal objectives functions via sampling different latent vectors from CVAE (right). By performing trajectory optimization with these two different objective functions, DiffTORI can generate multimodal actions (left).

different objective functions $f_\theta(z, a)$ and dynamics functions $d_\theta(z, a)$, allowing DiffTORI to generate different actions from the same state. Figure 4 illustrates the multimodal objective function learned by DiffTORI (right), and the resulting multimodal actions (left). The left subplot shows that when starting from the same action initialization $a_{init}$, with two different latent samples, DiffTORI optimizes $a_{init}$ into two different actions, $\hat{a}_1$ and $\hat{a}_2$ that move in distinct directions. The trajectory optimization procedure that iteratively updates the action is represented by dashed lines transitioning from faint to solid. From these two actions, two distinct trajectories are subsequently generated to push the T-shape object towards its goal. The middle and right subplots show the objective function landscapes for the 2 different samples, as well as the initial action $a_{init}$, and the final optimized action $\hat{a}_1$ and $\hat{a}_2$. We note the two landscapes are distinct from each other with different optimal solutions, showing that DiffTORI can generate multimodal objective functions and thus capture multimodal action distributions. We note that the learned objective function $f$ is not necessarily a "reward" function as those learned via inverse RL [48]. It is just a learned "objective function", such that optimizing it with trajectory optimization would yield actions that minimize the imitation learning loss with respect to the expert actions in the demonstration. We leave exploring the connections with inverse RL for future work.

### 5.2.3 ManiSkill

ManiSkill [20; 21] is a benchmark for learning generalizable robotic manipulation skills with 2D & 3D visual input. It includes a series of rigid body tasks and soft body tasks. We choose 9 tasks (4 soft body tasks and 5 rigid body tasks) from ManiSkill1 [20] and ManiSkill2 [21] and use 3D point cloud input for all the tasks. We use the end-effector frame as the observation frame [49] and use the PD controller with the end-effector delta pose as the action.

We build our method on top of the strongest imitation learning baseline in ManiSkill2 released by the authors, which is a Behavior Cloning (BC) policy with PointNet [50] as the encoder. Again, we also compare to BC+residual, which learns a residual policy that takes as input the action from the BC policy and outputs a delta correction. The results are shown in Table 3. As shown, DiffTORI + BC consistently achieves higher success rates than both baselines on all tasks, demonstrating the strong effectiveness of using differentiable trajectory optimization as the policy class.

## 6   Conclusion and Discussion

We introduce DiffTORI that uses differentiable trajectory optimization to generate the policy actions for deep reinforcement learning and imitation learning. The key is to utilize the recent progress in differentiable trajectory optimization to compute the gradients of the loss with respect to the parameters of the cost and dynamics function of trajectory optimization, and learn them end-to-end. When applied to model-based reinforcement learning, DiffTORI addresses the "objective mismatch" issue of prior methods. We also test DiffTORI for imitation learning on standard robotic manipulation task suites with high-dimensional sensory observations and compare it to feed-forward policy classes as well as Energy-Based Models (EBM) and Diffusion. Across 15 model-based RL tasks and 35 imitation learning tasks with high-dimensional image and point cloud inputs, DiffTORI outperforms prior state-of-the-art methods.

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

# A Additional results

## A.1 Model-based Reinforcement Learning

### A.1.1 DiffTORI without policy gradient loss

In model-based reinforcement learning, the key distinctions between DiffTORI and TD-MPC [38] are: 1) TD-MPC employs the Model Predictive Path Integral (MPPI [51]) in the planning stage, whereas we utilize trajectory optimization. 2) In addition to the original loss used in TD-MPC, we use an additional policy gradient loss and back-propagate it through the differentiable trajectory optimization process to update the model parameters. Figure 5 shows that the improvement of DiffTORI over TD-MPC comes from the addition of the policy gradient loss, instead of purely changing MPPI to trajectory optimization. To be more specific, we compare TD-MPC with DiffTORI (w/o backward), a variant of DiffTORI that removes the policy gradient loss for updating the model parameters. The results indicate that TD-MPC and the DiffTORI (w/o backward) variant perform comparably, suggesting that using MPPI or trajectory optimization at test-time for action generation have similar performances. With the inclusion of the policy gradient loss, DiffTORIsignificantly outperforms both TD-MPC and the DiffTORI (w/o backward) variant, demonstrating the efficacy of adding the policy gradient loss in DiffTORI.

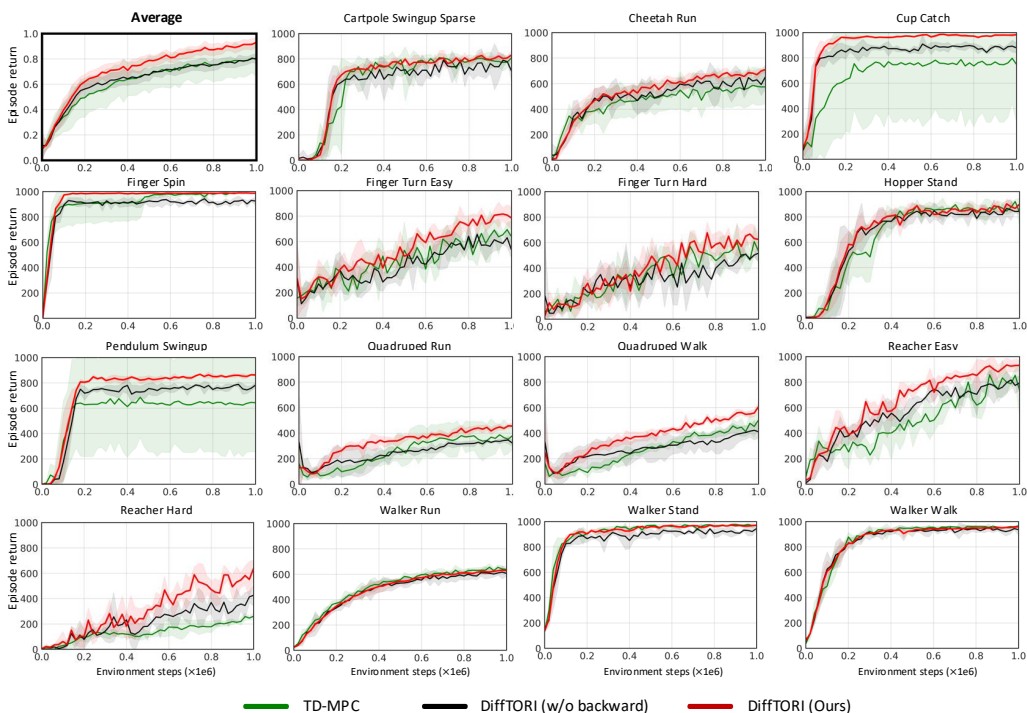

Figure 5: Performance of DiffTORI, in comparison to TD-MPC and DiffTORI (w/o backward) on 15 tasks from DeepMind control suite.

### A.1.2 Computational efficiency of DiffTORI

In addition to comparing the sample efficiency of DiffTORI to prior methods, we also compare the computational efficiency of DiffTORI versus TD-MPC on some of the environments. This is shown in Figure 7, where the y-axis is the return, and the x-axis is the wall-clock time (tested on a NVIDIA RTX 2080 Ti GPU) used to train DiffTORI and TD-MPC for 1M environment steps. As shown, it takes more wall-clock time for DiffTORI to finish the training. In terms of computational efficiency, the results are environment-dependent. DiffTORI achieves better computational efficiency on reacher-hard and cup-catch. On pendum-swingup, TD-MPC converges to a sub-optimal value in the early training stage and DiffTORI outperforms it within 24 hours of training time. DiffTORI

has similar computational efficiency on cartpole-swingup-sparse, reacher-easy, and finger-spin, and slightly worse computational efficiency on cheetah-run and walker-stand compared to TD-MPC. The gap is larger on hopper-stand. The major reason for DiffTORI to take longer time for training is that solving and back-propagating through the trajectory optimization problem in (4) is slower than doing MPPI as used in TD-MPC. As a reference, to infer the action at one time step, it takes 0.052 second to use Theseus to solve and differentiate through the trajectory optimization problem in (4), and 0.0092 second for using MPPI in TD-MPC. However, we also want to note that the community is actively developing better and faster algorithms/software libraries for differentiable trajectory optimization, which could improve the computation efficiency of DiffTORI. For example, in all our experiments, we used the default CPU-based solver in Theseus. Theseus also provides a more advanced solver named BaSpaCho, which is a batched sparse Cholesky solver with GPU support. When we switch from the default CPU-based solver to BaSpaCho, the time cost of solving the trajectory optimization problem in (4) is reduced by 22% from 0.052 second to 0.041 second. With better libraries/algorithms in the future for differentiable trajectory optimization, we believe the computational efficiency of DiffTORI would further improve.

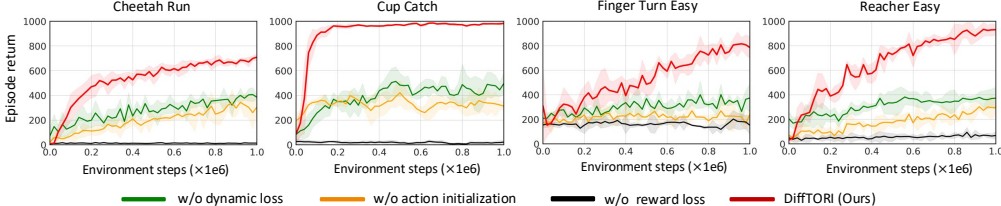

Figure 6: Ablation study of DiffTORI to examine the contribution of each loss terms towards the final performance, on a subset of 4 tasks. We find the reward prediction loss, action initialization, and dynamics prediction loss are all essential for DiffTORI to achieve good performance.

### A.1.3 Ablation study on the loss terms

We also perform ablation studies to examine how each loss term in (6) contributes to the final performance of DiffTORI, as shown in Figure 6. We find that removing the reward prediction loss causes DiffTORI to completely fail. Removing the dynamics loss, or not using the action initialization from the learned policy $\pi_\psi$ for solving the trajectory optimization, both lead to a decrease in the performance. These shows the necessity of using all the loss terms in DiffTORI for learning a good latent space to achieve strong performance.

## A.2 Imitation Learning

### A.2.1 DiffTORI with zero and random action initialization

We also present results of DiffTORI with zero initialization or random initialization, where instead of initializing the action from a base policy, the action is initialized to be 0, or randomly sampled from $\mathcal{N}(0, 1)$, on RoboMimic and Maniskill.

The results on RoboMimic is shown in Table 4. We notice a drop in performance of DiffTORI with zero or randomly-initialized actions, possibly due to the convergence to bad local minima during nonlinear trajectory optimization without a good action initialization. We note this would not be a drawback of applying DiffTORI in practice for imitation learning: one could always first learn a base policy using any behavior cloning algorithm, and then use DiffTORI to further refine the actions.

The results on Maniskill is shown in Table 5. Again, if we use zero or random action initialization with DiffTORI, the performance drops to be similar to or slightly worse than vanilla BC. Therefore, we think a good practice of using DiffTORI for imitation learning would be to always try to provide it with a good action initialization, e.g., by first training a BC policy and use its action as the initialization in DiffTORI.

### A.2.2 Results of positional controller on RoboMimic

Note that for the three tasks in Table 2 from Robomimic, we use the default velocity controller from Robomimic. We note the use of the velocity controller leads to a small decline in the performance of

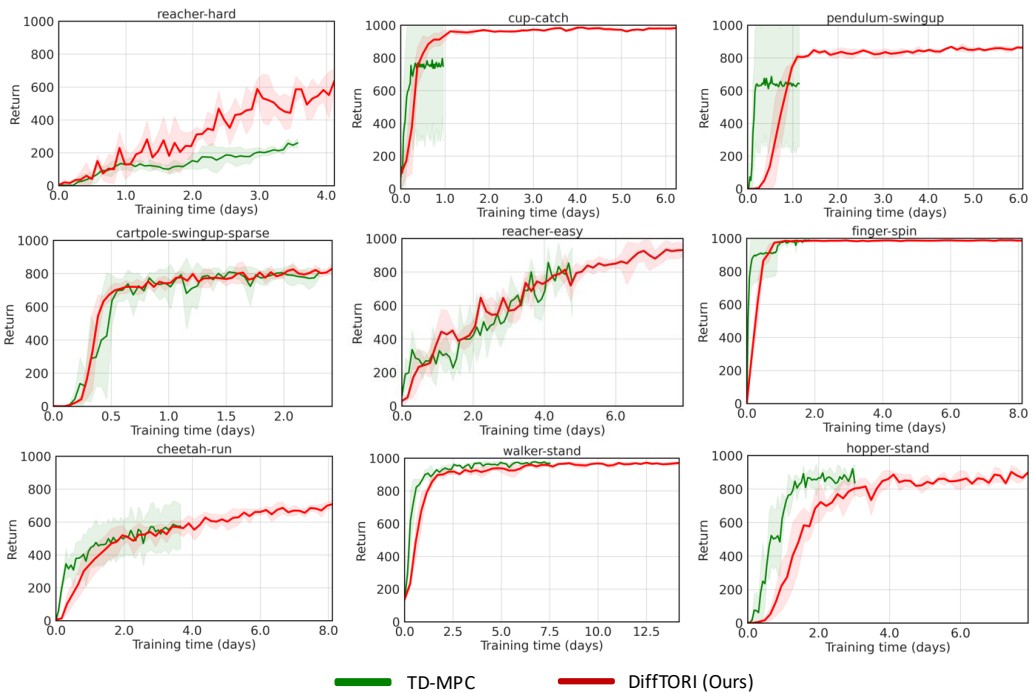

Figure 7: Return vs wall-clock time of DiffTORI and TD-MPC on some of the RL environments. The x-axis is the training time in days (24 hours), and the y-axis is the return. Both methods are trained for 1M environments steps. The training takes a long time (a few days on some environments) because the policy observation is high-dimensional images.

Table 4: Failure rates (↓) of DiffTORI and all other mehtods on the Robomimic tasks. DiffTORI achieves the best performances on all tasks when using diffusion policy as the base policy. If zero or random initialization are used in DiffTORI, the performance drops, possibly due to the convergence to bad local minima during nonlinear trajectory optimization without a good action initialization.

|  | IBC | BC-RNN | Residual +BC-RNN | DiffTORI (Ours) + BC-RNN | Diffusion | IBC + Diffusion | Residual + Diffusion | DiffTORI (Ours) + Diffusion | DiffTORI (Ours) + zero init. | DiffTORI (Ours) + random init. |
|---|---|---|---|---|---|---|---|---|---|---|
| Square | 0.96±0.00 | 0.18±0.00 | 0.16±0.01 | 0.10±0.02 | 0.12±0.03 | 0.32±0.05 | 0.12±0.02 | **0.08**±0.01 | 0.16±0.02 | 0.20±0.00 |
| Transport | 1.00±0.00 | 0.28±0.03 | 0.26±0.03 | 0.17±0.02 | 0.07±0.04 | 0.92±0.03 | 0.08±0.01 | **0.04**±0.01 | 0.58±0.01 | 0.64±0.04 |
| ToolHang | 1.00±0.00 | 0.33±0.04 | 0.28±0.03 | 0.18±0.00 | 0.10±0.00 | 0.94±0.01 | 0.10±0.00 | **0.08**±0.01 | 1.00±0.00 | 1.00±0.00 |
| Push-T | 0.89±0.01 | 0.30±0.02 | 0.28±0.02 | 0.25±0.02 | **0.09**±0.00 | 0.92±0.01 | **0.09**±0.00 | 0.09±0.01 | 0.38±0.04 | 0.43±0.02 |

the Diffusion Policy compared to its performance in the original paper where a positional controller is used. The Push-T task still uses the default position controller as in the diffusion policy paper. Below we evaluate the performance of DiffTORI and Diffusion Policy with the positional controller.

In the original Diffusion Policy [1] paper, it was observed that the use of positional controllers yielded superior results for Diffusion Policy compared to the default velocity controller on Robomimic [19] tasks. We evaluate Diffusion Policy, which is the strongest baseline, and DiffTORI on the most difficult three tasks with ph (proficient-human demonstration) and mh (multi-human demonstration) demonstrations using positional controller. The results with the positional controller are presented in Table 6. Diffusion Policy already achieves nearly the maximal possible performance on most tasks with the positional controller. DiffTORI, however, is able to achieve similar or even higher performances on most of these tasks.

### A.2.3  Ablation on planning horizon $H$

Additionally, we do ablation experiments on the planning horizon $H$ for imitation learning, with the results presented in Table 7. We observe that simply increasing the planning horizon $H$ in imitation learning does not necessarily enhance performance. As the horizon increases from $H = 1$ to $H = 3$,

Table 5: Success rates (↑) of all methods on the Maniskill benchmark. DiffTORI consistently outperforms both baselines on all tasks with action initialization from the BC policy. If zero or random initialization are used in DiffTORI, the performance drops, possibly due to the convergence to bad local minima during nonlinear trajectory optimization without a good action initialization.

| | PickCube | Fill | Hang | Excavate | Pour | OpenCabinet Drawer | OpenCabinet Door | PushChair | MoveBucket |
|---|---|---|---|---|---|---|---|---|---|
| BC | 0.19±0.03 | 0.72±0.04 | 0.76±0.02 | 0.25±0.02 | 0.13±0.01 | 0.47±0.03 | 0.35±0.04 | 0.12±0.01 | 0.10±0.01 |
| BC + residual | 0.21±0.04 | 0.75±0.02 | 0.75±0.02 | 0.27±0.03 | 0.12±0.01 | 0.49±0.02 | 0.36±0.03 | 0.15±0.02 | 0.10±0.01 |
| DiffTORI(Ours) + BC | **0.32**±0.02 | **0.82**±0.01 | **0.85**±0.03 | **0.29**±0.01 | **0.17**±0.02 | **0.53**±0.02 | **0.45**±0.02 | **0.20**±0.02 | **0.15**±0.02 |
| DiffTORI (Ours) + zero init. | 0.20±0.03 | 0.76±0.03 | 0.72±0.02 | 0.25±0.01 | 0.04±0.00 | 0.50±0.04 | 0.34±0.04 | 0.04±0.01 | 0.06±0.00 |
| DiffTORI (Ours) + random init. | 0.18±0.02 | 0.68±0.03 | 0.67±0.01 | 0.19±0.04 | 0.04±0.00 | 0.39±0.04 | 0.30±0.02 | 0.00±0.00 | 0.05±0.01 |

Table 6: Failure rates (↓) of DiffTORI and Diffusion Policy using Positional Controllers on Robomimic Tasks.

| | Square (ph) | Square (mh) | Transport (ph) | Transport (mh) | ToolHang (ph) |
|---|---|---|---|---|---|
| Diffusion | **0.02**±0.01 | **0.03**±0.02 | **0.00**±0.00 | 0.12±0.02 | 0.05±0.02 |
| DiffTORI + Diffusion | **0.02**±0.01 | 0.04±0.02 | **0.00**±0.00 | **0.09**±0.01 | **0.04**±0.01 |

the performance remains nearly the same; however, when $H$ is increase to 5, we observe a slight decline in the performance.

Table 7: Failure rates (↓) of different planning horizon $H$ for DiffTORI on RoboMimic tasks.

| | Square (ph) | Transport (ph) | ToolHang (ph) | Push-T |
|---|---|---|---|---|
| $H = 1$ | **0.08**±0.01 | **0.04**±0.01 | **0.08**±0.01 | **0.09**±0.01 |
| $H = 3$ | **0.08**±0.01 | 0.06±0.02 | **0.08**±0.00 | 0.12±0.02 |
| $H = 5$ | 0.09±0.01 | 0.06±0.01 | 0.10±0.00 | 0.12±0.01 |

### A.3  Comparison to prior work Amos et al. [3]

In our main paper, we did not compare to [3; 6; 8] because we target different experiments. These related works all conduct experiments on customized tasks with ground-truth low-level states. In contrast, we test our method on standard RL and robotic imitation learning benchmarks, with high-dimensional sensory observations like images and point clouds. As these prior works have not been demonstrated on high-dimensional observations or more complex tasks, we originally compared to more recent state-of-the-art methods on these benchmarks, e.g., 3D Diffusion Policy [46].

We have now included a comparison with Amos et al. [3] in one of their tasks (pendulum swing-up with ground-truth low-level states) under imitation learning settings. Unlike Amos et al. [3], who assumes known dynamics and reward structures and only learns 10 parameters, our method uses neural networks to represent both dynamics and reward functions without such assumptions. The metric is the cost of the learned policy. As in Amos et al., we test in two settings, pendulum without damping and with damping. Following Amos et al., their method does not model the damping effect in the assumed dynamics, so the ground-truth dynamics model is not realizable in the damping case. We also compared to an additional baseline in Amos et al., which uses a LSTM to predict the expert action. The results in Table 8 show our method performs slightly worse in the no damping case but noticeably better in the damping case. This is because Amos et al. assumes correct dynamics in the no damping case and learns only 10 unknown parameters, whereas the assumed dynamics structure is incorrect in the damping case; we use fully-connected neural networks to represent the dynamics function, avoiding such assumptions. It is generally difficult to know the exact correct dynamics function structure, especially for tasks with complex dynamics (e.g., with contacts) and high-dimensional observations (images and point clouds).

Table 8: Cost of different algorithms on the Pendulum swingup tasks from Amos et al. As in Amos et al., we test in two settings, pendulum without damping and with damping. Lower cost means the better performance. DiffTORI performs slightly worse in the no damping case but noticeably better in the damping case.

|  | Expert Policy | Amos et al. | LSTM policy | DiffTORI(ours) |
|---|---|---|---|---|
| Pendulum w/o damping | 13.126 | $13.576 \pm 0.012$ | $15.962 \pm 0.164$ | $14.603 \pm 0.190$ |
| Pendulum with dampling | 10.132 | $14.874 \pm 0.600$ | $12.098 \pm 0.031$ | $10.644 \pm 0.029$ |

## B  Implementation Details

In this section, we describe the implementation details of DiffTORI for the model-based RL experiments. For the imitation learning part, the code structure is very similar to this model-based RL implementation. For more detailed information, please refer to the code we will release upon acceptance of the paper. We implement DiffTORI on top of the open-source implementation of TD-MPC [38] from the authors. Below we show the pseudo-code of the training function in DiffTORI.

```python
def train():
    """
    Training code
    """
    for step in range(total_steps):
        obs = env.reset()
        # Differentiable trajectory optimization and update model
        action, info = agent.plan_theseus_update(obs)
        # Env step
        obs, reward, done, _ = env.step(action.cpu().numpy())
        # collect data in buffer and update model (TD-MPC loss)
        replay_buffer += (obs, action, reward, done)
        agent.update(replay_buffer)
```

Then, we demonstrate how the policy gradient loss is computed by differentiable trajectory optimization in DiffTORI with PyTorch-like pseudocode:

```python
def plan_theseus_update(obs):
    """
    Differentiable trajectory optimization and update model using policy
    gradient loss.
    h, R, Q, d: model components.
    c0: loss coefficients.
    """
    import theseus as th

    # Encode first observation
    z = self.model.h(obs)

    # Get initialization action from pi
    init_actions = self.model.pi(z)

    # Theseus variable
    actions = th.Vector(tensor=actions, name="actions")
    obs = th.Variable(obs, name="obs")

    # Cost Function and Objective
    cost_function = th.AutoDiffCostFunction([obs], [action]
        ,value_cost_fn)
    objective = th.Objective().add(cost_function)

    # Trajectory optimization optimizer
    theseus_optim = th.TheseusLayer(th_optimizer)

    # Theseus layer forward
```

```python
theseus_inputs = {"actions": init_actions, "obs": obs}
updated_inputs, info = theseus_optim.forward(theseus_inputs)
updated_actions = updated_inputs['actions']

# Update model using policy gradient losss
a_loss = - torch.min(*self.model.Q_s(obs, updated_actions[0]))*c0
a_loss.backward()
optim_a.step()
```

-For model-based reinforcement learning, We provide the network details for the added networks we used upon TD-MPC, which are the twin Q networks $\tilde{Q}_\phi$ learned in the original state space for computing the deterministic policy gradient.

```
( Q_s1 ):  Sequential (
    (0):  Linear ( in_features =S,  out_features =256)
    (1):  ELU( alpha =1.0)
    (2):  Linear ( in_features =256,  out_features =Z))
    (3):  Linear ( in_features =Z+A,  out_features =512)
    (4):  LayerNorm ((512 ,),  elementwise_affine =True )
    (5):  Tanh ()
    (6):  Linear ( in_features =512,  out_features =512)
    (7):  ELU( alpha =1.0)
    (8):  Linear ( in_features =512,  out_features =1))
( Q_s2 ):  Sequential (
    (0):  Linear ( in_features =S,  out_features =256)
    (1):  ELU( alpha =1.0)
    (2):  Linear ( in_features =256,  out_features =Z))
    (3):  Linear ( in_features =Z+A,  out_features =512)
    (4):  LayerNorm ((512 ,),  elementwise_affine =True )
    (5):  Tanh ()
    (6):  Linear ( in_features =512,  out_features =512)
    (7):  ELU( alpha =1.0)
    (8):  Linear ( in_features =512,  out_features =1))
```

For Imitation Learning, The default network details are as follows. Note that for Robomimic [19] and Push-T tasks, we use the RNN-encoder from Robomimic; for ManiSkill [20; 21] tasks, we use the PointNet encoder from ManiSkill2 [21].

```
( ho ):  Sequential (
    (0):  Linear ( in_features =S,  out_features =256)
    (1):  ELU( alpha =1.0)
    (2):  Linear ( in_features =256,  out_features =256)
    (3):  ELU( alpha =1.0)
    (4):  Linear ( in_features =256,  out_features =Zs ))
( ha ):  Identity
( hl ):  Sequential (
    (0):  Linear ( in_features =Zs+A,  out_features =256)
    (1):  ELU( alpha =1.0)
    (2):  Linear ( in_features =256,  out_features =256)
    (3):  ELU( alpha =1.0)
    (4):  Linear ( in_features =256,  out_features =128))
(R):  Sequential (
    (0):  Linear ( in_features =Zs+A+64,  out_features =512)
    (1):  ELU( alpha =1.0)
    (2):  Linear ( in_features =512,  out_features =512)
    (3):  ELU( alpha =1.0)
    (4):  Linear ( in_features =512,  out_features =1))
(d):  Sequential (
    (0):  Linear ( in_features =Zs+A+64,  out_features =512)
    (1):  ELU( alpha =1.0)
    (2):  Linear ( in_features =512,  out_features =512)
```

```
(3): ELU(alpha=1.0)
(4): Linear(in_features=512, out_features=Zs+64))
```

Hyperparameters used for DiffTORI for both model-based RL and imitation learning are shown in Tab 9. In model-based RL, we use the same parameters with TD-MPC [38] whenever possible.

Table 9: Hyperparameters used in DiffTORI.

| Hyperparameter | Value |
|---|---|
| Model-based RL | |
| Max planning iterations | 100 (50) |
| Planning step size | 1e-4 (5e-3) |
| Discount factor | 0.99 |
| Action loss coefficient (c0) | 1 |
| optimizer | Adam($\beta_1 = 0.9$, $\beta_2 = 0.999$) |
| Gradient Norm | 10 |
| Planning horizon schedule | $1 \rightarrow 5$ (25k steps) |
| Batch size | 256 |
| Latent dimension | 50 |
| Sampling technique | PER($\alpha = 0.6$, $\beta = 0.4$) |
| Learning rate | 1e-3 |
| Imitation Learning | |
| Max planning iterations | 100 |
| Planning step size | 1e-4 |
| Planning horizon schedule | 1 |
| Latent dimension | 50 |
| Posterior Gaussian dimension | 64 |
| KL coefficien | 1 |
| Learning rate | 3e-4 |
| Learning rate (MetaWorld) | 3e-3 |
| GMM Num Modes | 5 |
| RNN Seq Len | 16 |
| RNN Hidden Dim | 1000 |
| Point Cloud Sampled Points (ManiSkill) | 1200 |
| Point Cloud Sampled Points (MetaWorld) | 512 |

## C   Environment Details

For model-based reinforcement learning evaluation, we use 15 visual continuous control tasks from Deepmind Control Suite (DMC). For imitation learning, we use 13 tasks (detailed information can be found in Table 10) from Robomimic [19], IBC [2], ManiSkillp [20], and ManiSkill2 [21].

We visualize the keyframes of the imitation learning tasks in Fig 8 and Fig 9.

## D   More implementation details on using CVAE for imitation learning

We provide more details on how we instantiate DiffTORI with CVAE in imitation learning, in which the goal is to reconstruct the expert actions conditioned on the state. The CVAE encoder is composed of three networks: the first network is a state encoder $h_\theta^o$ that encodes the state into a latent feature vector $z^s = h_\theta^o(s_i)$, which is the conditional information in our case. The second is an action encoder $h_\theta^a$ that encodes the expert action into a latent feature vector $z^a = h_\theta^a(a_i^*)$. The last is a fusing encoder $h_\theta^l(z^s, z^a)$ that takes as input the concatenation of the state and action latent features, and outputs the mean $\mu$ and variance $\sigma^2$ of the posterior Gaussian distribution $\mathcal{N}(\cdot|\mu, \sigma^2)$. During training, the final latent state $z$ for state $s_i$ used in (7) is the concatenation of a sampled vector $\tilde{z}$ from the posterior Gaussian distribution $\mathcal{N}(\cdot|\mu, \sigma^2)$, and the latent state feature vector $z^s$: $z = [\tilde{z}, z^s], \tilde{z} \sim \mathcal{N}(\cdot|\mu, \sigma^2)$.

Table 10: Imitation Learning Tasks Summary.

| Task | Source | Obs. Type | Ac Dim | Object | Demo | Task Description |
|---|---|---|---|---|---|---|
| Square | Robomimic | Img | 7 | Rigid | 200 | Pick a square nut and place it on a rod. |
| Transport | Robomimic | Img | 14 | Rigid | 200 | Transfer a hammer from a container to a bin |
| ToolHang | Robomimic | Img | 7 | Rigid | 200 | assemble a frame consisting of a base and hook |
| Push-T | IBC | Img | 2 | Rigid | 200 | Push a T-shaped object to a specified position |
| OpenCabinetDrawer | ManiSkill1 | Point Cloud | 13 | Rigid | 300/obj. | Open a specific drawer of the cabinet |
| OpenCabinetDoor | ManiSkill1 | Point Cloud | 13 | Rigid | 300/obj. | Open a specific door of the cabinet |
| PushChair | ManiSkill1 | Point Cloud | 22 | Rigid | 300/obj. | Push the swivel chair to the target position |
| MoveBucket | ManiSkill1 | Point Cloud | 22 | Rigid | 300/obj. | Move a bucket and lift it onto a platform |
| PickCube | ManiSkill2 | Point Cloud | 7 | Rigid | 1000 | Pick up a cube and move it to a goal position |
| Fill | ManiSkill2 | Point Cloud | 7 | Soft | 200 | Fill clay from a bucket into the target beaker |
| Hang | ManiSkill2 | Point Cloud | 7 | Soft | 200 | Hang a noodle on a target rod |
| Excavate | ManiSkill2 | Point Cloud | 7 | Soft | 200 | Lift a amount of clay to a target height |
| Pour | ManiSkill2 | Point Cloud | 7 | Soft | 200 | Pour liquid from a bottle into a beaker |
| Soccer | MetaWorld (medium) | Point Cloud | 4 | Rigid | 10 | Kick a soccer into the goal |
| Push Wall | MetaWorld (medium) | Point Cloud | 4 | Rigid | 10 | Bypass a wall and push a puck to a goal |
| Peg insert side | MetaWorld (medium) | Point Cloud | 4 | Rigid | 10 | Insert a peg sideways |
| Bin picking | MetaWorld (medium) | Point Cloud | 4 | Rigid | 10 | Grasp the puck from one bin and place it into another bin |
| Basketball | MetaWorld (medium) | Point Cloud | 4 | Rigid | 10 | Dunk the basketball into the basket |
| Box close | MetaWorld (medium) | Point Cloud | 4 | Rigid | 10 | Grasp the cover and close the box with it |
| Coffee pull | MetaWorld (medium) | Point Cloud | 4 | Rigid | 10 | Pull a mug from a coffee machine |
| Coffee push | MetaWorld (medium) | Point Cloud | 4 | Rigid | 10 | Push a mug under a coffee machine |
| Hammer | MetaWorld (medium) | Point Cloud | 4 | Rigid | 10 | Hammer a screw on the wall |
| Sweep | MetaWorld (medium) | Point Cloud | 4 | Rigid | 10 | Sweep a puck off the table |
| Sweep into | MetaWorld (medium) | Point Cloud | 4 | Rigid | 10 | Sweep a puck into a hole |
| Assemble | MetaWorld (hard) | Point Cloud | 4 | Rigid | 10 | Pick up a nut and place it onto a peg |
| Hand insert | MetaWorld (hard) | Point Cloud | 4 | Rigid | 10 | Insert the gripper into a hole |
| Pick out of hole | MetaWorld (hard) | Point Cloud | 4 | Rigid | 10 | Pick up a puck from a hole |
| Pick place | MetaWorld (hard) | Point Cloud | 4 | Rigid | 10 | Pick and place a puck to a goal |
| Push | MetaWorld (hard) | Point Cloud | 4 | Rigid | 10 | Push the puck to a goal |
| Push back | MetaWorld (hard) | Point Cloud | 4 | Rigid | 10 | Pull a puck to a goal |
| Shelf place | MetaWorld (very hard) | Point Cloud | 4 | Rigid | 10 | pick and place a puck onto a shelf |
| Disassemble | MetaWorld (very hard) | Point Cloud | 4 | Rigid | 10 | pick a nut out of the a peg |
| Stick pull | MetaWorld (very hard) | Point Cloud | 4 | Rigid | 10 | Grasp a stick and pull a box with the stick |
| Stick push | MetaWorld (very hard) | Point Cloud | 4 | Rigid | 10 | Grasp a stick and push a box using the stick |
| Pick place wall | MetaWorld (very hard) | Point Cloud | 4 | Rigid | 10 | Pick a puck,bypass a wall and place the puck |

The latent state $z$ will then be used as input for the decoder, which consists of the reward function $R_\theta$, and the dynamics function $d_\theta$. Trajectory optimization is performed with the reward and dynamics function in the decoder to solve (7) to generate the reconstructed action $a^*(\theta; s_i)$. The loss for training the CVAE is the evidence lower bound (ELBO) on the demonstration data:

$$\mathcal{L}_{DiffTORI}^{IL}(\theta) = \sum_{i=1}^{N} ||a(\theta; s_i) - a_i^*||_2^2 - \beta \cdot \text{KL}(\mathcal{N}(\cdot|\mu, \sigma^2)|\mathcal{N}(0, I)), \qquad (9)$$

where $\text{KL}(P||Q)$ denotes the KL divergence between distributions $P$ and $Q$. At test time, only the decoder of the CVAE is used for generating the actions. Given a state $s$, the latent state $z$ is the concatenation of the encoded latent state feature $z^s$, and a sampled vector $\tilde{z}$ from the prior distribution $\mathcal{N}(0, 1)$.

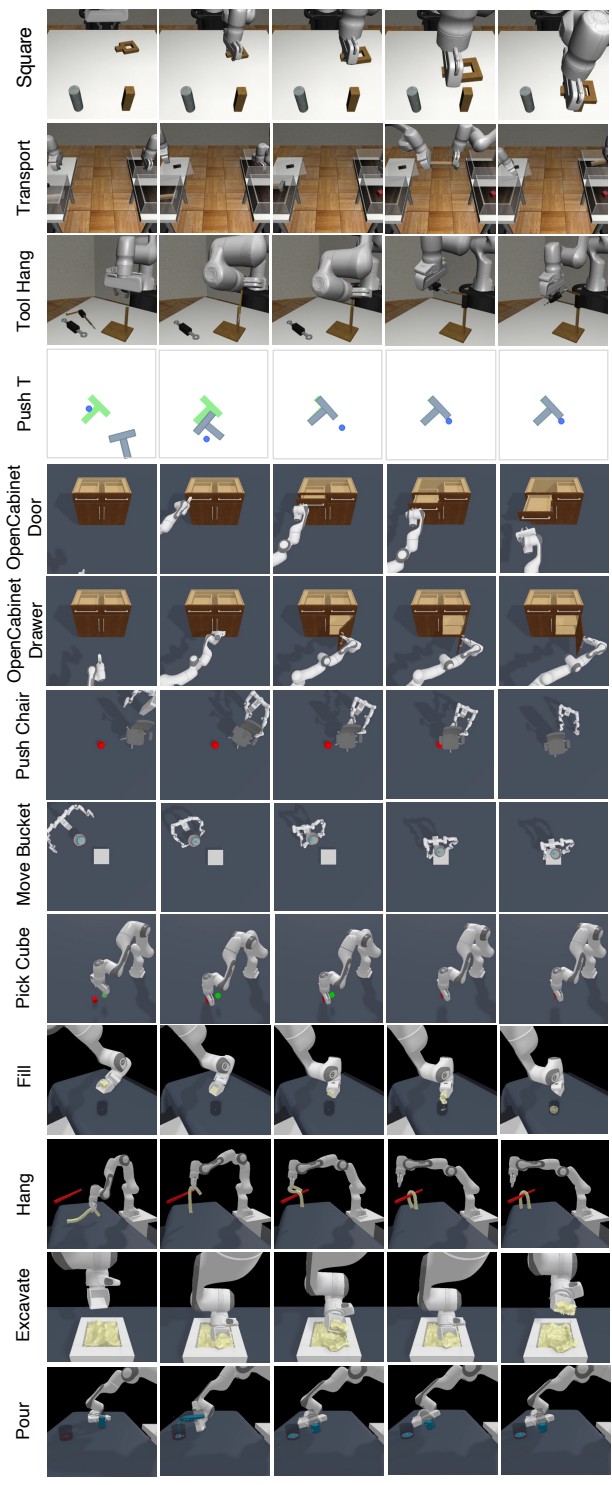

Figure 8: Visualization of the tasks for imitation learning in RoboMimic and ManiSkill.

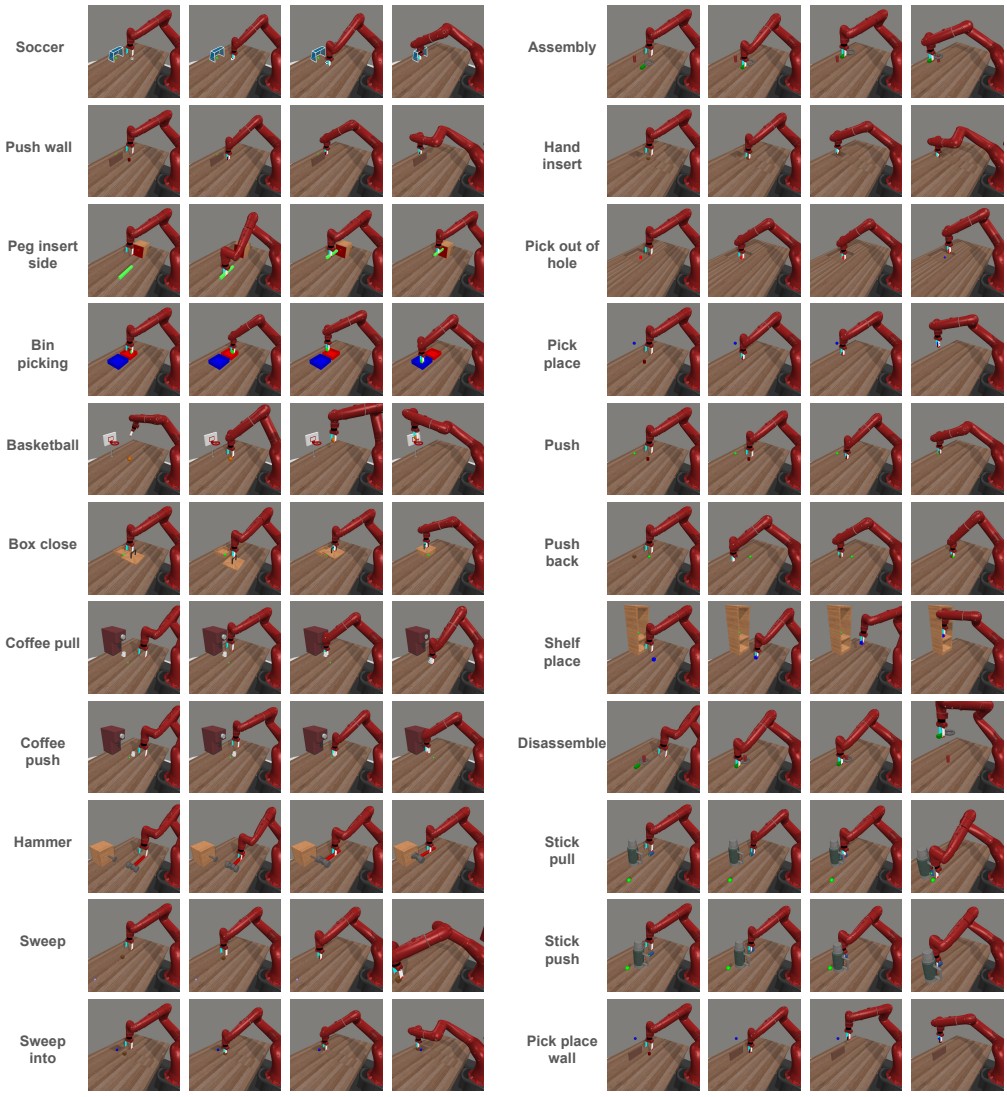

Figure 9: Visualization of the tasks for imitation learning in Metaworld.

