# OpenReview forum: "DiffTORI: Differentiable Trajectory Optimization for Deep Reinforcement and Imitation Learning"
_NeurIPS.cc/2024/Conference — NeurIPS 2024 spotlight_

### Official Review · Reviewer_XFZZ · 2024-06-30

**Soundness:** 3
**Presentation:** 3
**Contribution:** 3
**Rating:** 6
**Confidence:** 4

**Summary:**

This paper proposes DiffTOP as a model-based approach to reinforcement learning and behavior cloning. DiffTOP learns a cost function and a dynamics model through differentiable trajectory optimization, and then uses the learned model during inference for online optimization. For model-based RL tasks, DiffTOP is built on a recent work of TD-MPC and resolves the objective mismatch issue that TD-MPC failed to address. For behavior cloning, the DiffTOP can be used with a different loss function tailed to BC, with an additional consideration of learning a multimodal policy which is achieved by the use of CVAE. The proposed DiffTOP framework is evaluated on 15 model-based RL tasks and 35 imitation learning tasks with high-dimensional observation spaces.

**Strengths:**

* The authors demonstrate broad applicability of the proposed DiffTOP framework in both reinforcement learning and behavior cloning settings, with varying problem complexity ranging from simple continuous control (such as cartpole) to complex object manipulation tasks (such as robomimic or maniskill).

**Weaknesses:**

[EDIT] The concerns were resolved throughout the rebuttal. In particular, my initial assessment of the paper regarding the ignorance of TD-MPC2 was inaccurate.

Despite the thorough results presented in the paper, I have to say that it is not well contextualized relative to the existing literature.

* First and foremost, the idea of leveraging model-based differentiable optimization for reinforcement learning and learning-based control is not new, and the field is rapidly expanding with new methods emerging in multiple domains, including control theory, robotics, and machine learning. For instance, Nikishin et al. [1] proposes an approach to differentiable optimization of trajectories (for an infinite horizon) by leveraging implicit function theorem and deep neural networks. Cheng et al. [2][3] perform differentiable trajectory optimization to learn a feedback policy, although they assume a known model-based function class for the dynamics and the policy. Sacks et al. [4] propose to optimize the inner-loop of an MPC optimization algorithm, though their method is based on MPPI. Given the abundance of similar ideas in the literature, the authors are encouraged to spend time to conduct a more thorough literature review.

* Most critically, the authors of TD-MPC recently published a new paper titled TD-MPC2 [5] at ICLR 2024, in which the improvement from TD-MPC includes the mitigation of objective mismatch. Since the motivation is very similar to that of this paper and the authors of this paper completely disregard the new work, the contribution of this paper is highly questionable (even though the algorithmic details are still different between TD-MPC2 and DiffTOP). In particular, the following motivational statement made in Section 4 is no longer true: “Existing model-based RL algorithms such as TD-MPC suffer from the objective mismatch issue … DifftTOP addresses this issue …”

[1] Nikishin, Evgenii, Romina Abachi, Rishabh Agarwal, and Pierre-Luc Bacon. "Control-oriented model-based reinforcement learning with implicit differentiation." In Proceedings of the AAAI Conference on Artificial Intelligence, vol. 36, no. 7, pp. 7886-7894. 2022.

[2] Cheng, Sheng, Minkyung Kim, Lin Song, Chengyu Yang, Yiquan Jin, Shenlong Wang, and Naira Hovakimyan. "Difftune: Auto-tuning through auto-differentiation." arXiv preprint arXiv:2209.10021 (2022).

[3] Cheng, Sheng, Lin Song, Minkyung Kim, Shenlong Wang, and Naira Hovakimyan. "DiffTune $^+ $: Hyperparameter-Free Auto-Tuning using Auto-Differentiation." In Learning for Dynamics and Control Conference, pp. 170-183. PMLR, 2023.

[4] Sacks, Jacob, Rwik Rana, Kevin Huang, Alex Spitzer, Guanya Shi, and Byron Boots. "Deep model predictive optimization." arXiv preprint arXiv:2310.04590 (2023).

[5] Hansen, Nicklas, Hao Su, and Xiaolong Wang. "TD-MPC2: Scalable, Robust World Models for Continuous Control." In The Twelfth International Conference on Learning Representations (2024).

**Questions:**

* Regarding the use of CVAE for capturing multimodal action distributions for behavioral cloning, it seems that the distribution representation in the latent space is still unimodal Gaussian. Although it should work just fine, I wonder if the authors have considered to leverage a discrete categorical latent variable for explicitly modeling multi-modality even in the latent space to possibly improve the performance (see e.g., [6])

[6] Ivanovic, B. and Pavone, M., 2019. The trajectron: Probabilistic multi-agent trajectory modeling with dynamic spatiotemporal graphs. In Proceedings of the IEEE/CVF international conference on computer vision (pp. 2375-2384).

**Limitations:**

[EDIT] The concerns were resolved throughout the rebuttal. In particular, my initial assessment of the paper regarding the ignorance of TD-MPC2 was inaccurate.

* As mentioned above, the contribution of this paper is quite limited especially in light of the recent development of TD-MPC2.

---

> ### Author Rebuttal · Authors · 2024-08-06
>
> We want to extend our heartfelt gratitude for taking the time to review our paper. Thank you for all valuable comments and suggestions on improving the quality of the paper. Below we respond to each of your comments in detail.
>
> **Q: ... Given the abundance of similar ideas in the literature, the authors are encouraged to spend time to conduct a more thorough literature review.**
>
> We thank the reviewer for bringing up these related works. We agree that the field of using differentiable optimization for end-to-end learning is rapidly expanding, and we have already discussed and cited many relevant works in the “Differentiable optimization” paragraph in the related work section. We are sorry for the omission of these 4 related works and we have updated section 2 of our paper to include a thorough discussion of them, detailed as follows:
>
> Nikishin et al. [1] proposes to learn a dynamics and reward model in model-based RL, and they derive an implicit policy as the softmax policy associated with the optimal Q function under the learned dynamics and reward. The dynamics and reward model are learned by back-propagating the RL loss through this implicit policy via implicit function theorem. In contrast, we derive the implicit policy as the optimal solution from performing trajectory optimization with the learned dynamics, reward and Q function. Nikishin et al. only tested their methods in simple tasks with ground-truth low-level states (e.g., CartPole with only 2d action space and 4d state space), while we show our methods work with much more complex tasks in DeepMind Control Suite with high-dimensional image observations, and it outperforms several prior state-of-the-art methods.
>
> Cheng et al. [2, 3] proposes to tune the parameters of a controller by unrolling the controller and system dynamics into a computation graph and optimizing the controller parameters via gradient descent with respect to the task loss, with applications in tracking trajectories for drones. The method assumes a known dynamics and policy class (e.g., a PID controller), and only optimizes some parameters of the controller. Our method does not assume any prior knowledge on the dynamics or policy class, instead, the dynamics and policy are all neural networks which we learn end-to-end using the task loss. Instead of representing the policy as a predefined controller and learning its parameters, our policy is represented as performing trajectory optimization with the learned dynamics, reward and Q functions.
>
> Sacks et al. [4] proposes to learn the update rule in MPPI (the mean and variance used to sample the actions), represented as a neural network, using reinforcement learning, with known dynamics and cost functions. To apply RL for learning the update rule, they design an auxiliary MDP and treat the MPPI process as part of the dynamics of this auxiliary MDP. Instead of learning the update rule, we learn the dynamics, reward, Q function used in trajectory optimization to generate the actions. We perform differentiable trajectory optimization instead of RL to optimize the parameters of these functions.
>
> We thank the reviewer again for introducing us to these 4 related works. As discussed above, we believe our proposed method is quite different from these 4 related works and our contributions remain valid.
>
> **Q: The authors of TD-MPC recently published a new paper titled TD-MPC2 [5] at ICLR 2024, in which the improvement from TD-MPC includes the mitigation of objective mismatch. Since the motivation is very similar to that of this paper and the authors of this paper completely disregard the new work, the contribution of this paper is highly questionable.**
>
> We thank the reviewer for raising this issue. We respectfully disagree with this assessment. TD-MPC2 [5] does not include any improvements over TD-MPC [6] that mitigate the objective mismatch issue. TD-MPC2 inherits the same training objective from TD-MPC, as evidenced by Equation (3) in TD-MPC2 and Equations (8), (9), and (10) in TD-MPC. This means that TD-MPC2 still suffers from the objective mismatch issue, where the dynamics model is optimized to predict future states in latent space (via the latent state consistency loss), which is not necessarily aligned with the goal of achieving high task return when using the dynamics model for planning. The primary improvements in TD-MPC2 over TD-MPC are related to exploring different architectural variations and new algorithmic design choices for more stable training in multi-task settings, rather than addressing the objective mismatch issue. Therefore, we believe our contribution and motivation remains valid. We have updated section 2 of our paper to include a discussion of TD-MPC2.
>
> **Q: I wonder if the authors have considered to leverage a discrete categorical latent variable for explicitly modeling multi-modality even in the latent space to possibly improve the performance**
>
> Thank you for this great suggestion. We chose CVAE due to its simplicity, and as shown in our experiments, it achieves good performance and is able to capture the multi-modality well (Fig. 4). We believe that a more advanced technique, such as the discrete categorical latent variable as suggested, would further improve the performance. We look forward to exploring this in future work.
>
>
> [1] Nikishin et al., Control-oriented model-based reinforcement learning with implicit differentiation, AAAI Conference on Artificial Intelligence. 2022.
> [2] Cheng et al., Difftune: Auto-tuning through auto-differentiation, IEEE Transactions on Robotics, 2024
> [3] Cheng et al., DiffTune$^\dagger$: Hyperparameter-Free Auto-Tuning using Auto-Differentiation, Learning for Dynamics and Control Conference, 2023.
> [4] Sacks et al., Deep model predictive optimization, arXiv preprint, 2023.
> [5] Hansen et al., TD-MPC2: Scalable, Robust World Models for Continuous Control, ICLR 2024
> [6] Hansen et al., Temporal Difference Learning for Model Predictive Control, ICML 2022

---

> > ### Comment · Reviewer_XFZZ · 2024-08-09
> > **Response to Authors and Acknowledgement of Inaccurate Feedback in Initial Review**
> >
> > I sincerely thank the authors for their time and effort to prepare the rebuttal. Please find below my response.
> >
> > * > We thank the reviewer for bringing up these related works. We agree that the field of using differentiable optimization for end-to-end learning is rapidly expanding, and we have already discussed and cited many relevant works in the “Differentiable optimization” paragraph in the related work section. We are sorry for the omission of these 4 related works and we have updated section 2 of our paper to include a thorough discussion of them, detailed as follows:
> >
> >   * Thank you for conducting the thorough literature review. It should further clarify the uniqueness of DiffTOP when compared against prior approaches that share similarity.
> >
> >
> > * > We thank the reviewer for raising this issue. We respectfully disagree with this assessment. TD-MPC2 [5] does not include any improvements over TD-MPC [6] that mitigate the objective mismatch issue. TD-MPC2 inherits the same training objective from TD-MPC, as evidenced by Equation (3) in TD-MPC2 and Equations (8), (9), and (10) in TD-MPC. This means that TD-MPC2 still suffers from the objective mismatch issue, where the dynamics model is optimized to predict future states in latent space (via the latent state consistency loss), which is not necessarily aligned with the goal of achieving high task return when using the dynamics model for planning. The primary improvements in TD-MPC2 over TD-MPC are related to exploring different architectural variations and new algorithmic design choices for more stable training in multi-task settings, rather than addressing the objective mismatch issue. Therefore, we believe our contribution and motivation remains valid. We have updated section 2 of our paper to include a discussion of TD-MPC2.
> >
> >   * Thank you for your response. I appreciate the level of details provided in the exposition and apologize for mistakenly making an inaccurate claim in my original review; I had been confused in part by the following statement made in Section 3.1 of TD-MPC2 paper, which I cite below for transparency:
> >     * > However, accurately predicting raw future observations (e.g., images or proprioceptive features) over long time horizons is a difficult problem, and does not necessarily lead to effective control (Lambert et al., 2020). Rather than explicitly modeling dynamics using reconstruction, TD-MPC2 aims to learn a maximally useful model: a model that accurately predicts outcomes (returns) conditioned on a sequence of actions.
> >
> >     Here, Lambert et al. (2020) is the original objective mismatch paper. It implicitly states that TD-MPC2 takes care of the objective mismatch issue. However, based on your response (and upon closer look at Appendix A of TD-MPC2), I agree that the model objective of TD-MPC2 is largely the same as that of TD-MPC. Thus, the results presented in Figure 3 of the present paper already confirms the superiority of DiffTOP over TD-MPC/TD-MPC2 in terms of directly optimizing the task performance. In the revised paper, the authors may want to explicitly counter-argue the above statement made by the TD-MPC2 authors, rather than just stating that TD-MPC suffers from the objective mismatch issue (e.g., in Section 4.1), since these two statements are indeed conflicting with each other and may cause confusion for other people as well.
> >
> > Again, I appreciate the authors for taking the review comments seriously. I have updated my review and revised the score.

---

> ### Author Response · Authors · 2024-08-09
> **Thank you for your response!**
>
> We want to sincerely thank the reviewer for the prompt response, and the detailed explanation. We are glad our rebuttal has addressed your concerns, and we really appreciate the update of the review and the revision of the score.
>
> > It should further clarify the uniqueness of DiffTOP when compared against prior approaches that share similarity.
>
> Thank you for this suggestion! In addition to the discussion to the 4 related works in the initial rebuttal, we will also update the paper to further clarify the uniqueness of our method when compared to prior approaches.
>
> We want to thank the reviewer again for taking the time to review our paper and read our rebuttal. Your suggestions and feedback have greatly helped improve the quality of the paper.

---

### Official Review · Reviewer_epML · 2024-07-09

**Soundness:** 3
**Presentation:** 2
**Contribution:** 2
**Rating:** 5
**Confidence:** 4

**Summary:**

The paper introduces DiffTOP, an approach leveraging differentiable trajectory optimization as the policy representation to enhance performance in deep reinforcement learning (RL) and imitation learning (IL). By utilizing the advancements in differentiable trajectory optimization, DiffTOP addresses the "objective mismatch" issue prevalent in prior model-based RL algorithms by optimizing the dynamics model to directly maximize task performance. The method is benchmarked across various robotic manipulation tasks with high-dimensional sensory inputs and demonstrates superior performance over state-of-the-art methods in both model-based RL and IL domains.

**Strengths:**

1. The paper is well-written, structured, and intuitive, making it easy to understand.
2. The experiments are well-executed with various benchmarks. The comparison with baselines is thorough, including state-of-the-art methods like DP3.
3. The paper provides experiments in both imitation learning and reinforcement learning settings, which is commendable.
4. DiffTOP is combined with various existing approaches to demonstrate the generalization of the method.

**Weaknesses:**

1. The work is not well-contextualized. While the paper makes a general claim about a policy class using differentiable trajectory optimization, using such technique is not new in the field, as the authors point out in the related work section. Despite the authors mention some differences in implementation (e.g., whether the dynamics model is learned), the paper presents this differentiable trajectory optimization as a new contribution. It would be beneficial for the introduction and diagrams to highlight the specific differences between the proposed work and other differentiable trajectory optimization approaches, rather than focusing on the differences among EBM Diffusion policy and TD-MPC. Additionally, calling the method Differentiable Trajectory Optimization might be inaccurate since it encompasses a broad range of works and could be misleading, failing to capture the differences.
2. While the execution is commendable, the design choices could be better justified, and more analysis would strengthen the paper. Particularly, as mentioned in the related work section, there are closely related works like [2][19][42], but there is no direct comparison with them in the experiments. More analysis on the design choices, such as using TD-MPC for model-based RL and learning dynamic functions, would also be appreciated.
3. The idea is not particularly novel. The combination of TD-MPC and differentiable trajectory optimization is rather straightforward.
4. It would also be beneficial to have more validation on real robots.

**Questions:**

see the weaknesses section.

**Limitations:**

yes

---

> ### Author Rebuttal · Authors · 2024-08-06
>
> We want to extend our heartfelt gratitude for taking the time to review our paper. Thank you for all valuable suggestions and comments on improving the quality of the paper. Below we respond to each of your comments in detail.
>
> **Q:  The work is not well-contextualized … It would be beneficial for the introduction and diagrams to highlight the specific differences between the proposed work and other differentiable trajectory optimization approaches …  calling the method Differentiable Trajectory Optimization might be inaccurate since it encompasses a broad range of works and could be misleading, failing to capture the differences.**
>
> Thank you for this suggestion. We have updated the introduction and diagrams of our paper to highlight the differences between our method and other differentiable trajectory optimization approaches: “1) We are the first to show how differentiable trajectory optimization can be combined with deep model based RL algorithms, training dynamics, reward, Q function, and the policy end-to-end using task loss. In contrast, prior work focuses on imitation learning [2, 42], assumes known dynamics and reward structures and learns only a few parameters [2], or first learns the dynamics model with the dynamics prediction loss (instead of the task loss), and then uses the fixed learned dynamics for control [19]. 2) We are the first to show that the policy class by differentiable trajectory optimization can scale up to high-dimensional sensory observations like images and point clouds, achieving state-of-the-art performances in standard RL and imitation learning benchmarks. In contrast, prior works [2, 19, 42] only test their methods in customized tasks with ground-truth low-level states, such as CartPole or Pendulum, and do not report performance on standard benchmarks with more complex tasks and high-dimensional observations. ”
>
> Thank you for the suggestion on the name of the method, and we will revise it to better reflect our unique contributions.
>
> **Q: ... there is no direct comparison with closely related works [2][19][42] in the experiments.**
>
> Thank you for the feedback. We did not compare to [2][19][42] because we target different experiments. These related works all conduct experiments on customized tasks with ground-truth low-level states. In contrast, we test our method on standard RL and robotic imitation learning benchmarks, with high-dimensional sensory observations like images and point clouds. As these prior works have not been demonstrated on high-dimensional observations or more complex tasks, we originally compared to more recent state-of-the-art methods on these benchmarks, e.g., 3D Diffusion Policy [1].
>
> We have now included a comparison with Amos et al. in one of their tasks (pendulum swing-up with ground-truth low-level states) under imitation learning settings. Unlike Amos et al., who assumes known dynamics and reward structures and only learns 10 parameters, our method uses neural networks to represent both dynamics and reward functions without such assumptions. The metric is the cost of the learned policy. As in Amos et al., we test in two settings, pendulum without damping and with damping. Following Amos et al., their method does not model the damping effect in the assumed dynamics, so the ground-truth dynamics model is not realizable in the damping case. We also compared to an additional baseline in Amos et al., which uses a LSTM to predict the expert action.
>
> The results (Table 1 in the PDF uploaded via global rebuttal) show our method performs slightly worse in the no damping case but noticeably better in the damping case. This is because Amos et al. assumes correct dynamics in the no damping case and learns only 10 unknown parameters, whereas the assumed dynamics structure is incorrect in the damping case; we use fully-connected neural networks to represent the dynamics function, avoiding such assumptions. It is generally difficult to know the exact correct dynamics function structure, especially for tasks with complex dynamics (e.g., with contacts) and high-dimensional observations (images and point clouds).
>
> **Q: More analysis on the design choices, such as using TD-MPC for model-based RL and learning dynamic functions, would also be appreciated.**
>
> Thank you for the suggestion. We have updated section 5 of our paper to include more discussion and experiments on these design choices:
> “We choose TD-MPC for its simplicity and state-of-the-art performance. However, our method is compatible with any model-based RL algorithm that learns a dynamics model and a reward function. To show this, we have added experiments implementing our method on top of Dreamer-V3 [3], another state-of-the-art image-based model-based RL algorithm, and tested it on 4 Deepmind Control Suite tasks. The results (Fig. 1 of the PDF uploaded in the global rebuttal) show that integrating our method with Dreamer-V3 improves performance in 3 out of 4 tasks, indicating that our method can enhance other model-based RL algorithms as well. In our experiments, we have also combined our method with 3D Diffusion Policy for imitation learning, and it achieves significant improvements.
>
> We choose to learn the dynamics function as we work directly with high-dimensional sensory inputs like images and point clouds, where manually specifying the analytic dynamics function is challenging. This is in contrast to learning from ground-truth low-level states where the dynamics model can be derived using physical laws. Therefore, we need to learn a dynamics model in the latent space, similar to prior work such as TD-MPC [4]. ”
>
> **Q: It would also be beneficial to have more validation on real robots.**
>
> Thank you for the suggestion. While real-world validation is beyond the scope of this paper, which focuses on standard benchmarks in simulation, we believe that validation of our method on real robots would be valuable for the robotics community. we leave this as important future work.

---

> > ### Comment · Reviewer_epML · 2024-08-09
> >
> > Thank you for the detailed and well-written response! I particularly appreciate the authors' effort in conducting additional experiments within such a short period. Overall, I remain positive about the paper. One suggestion I would like to make is that DiffTop should be clearly distinguished from the previous works mentioned in the related work section. Currently, the paper’s organization doesn’t make this distinction immediately clear. A dedicated section, perhaps with a simple diagram highlighting the conceptual differences, would be helpful.

---

> ### Author Response · Authors · 2024-08-06
> **References for the rebuttal**
>
> [1] Ze, Y., et al.,  3d diffusion policy: Generalizable visuomotor policy learning via simple 3d representations, RSS, 2024
> [2] Amos, B., et al., Differentiable mpc for end-to-end planning and control. NeurIPS, 2018
> [3] Hafner, D., et al., Mastering diverse domains through world models. arXiv preprint arXiv:2301.04104, 2013
> [4] Hansen N., et al., Temporal Difference Learning for Model Predictive Control, ICML 2022
> [19] Jin, W., et al., Pontryagin differentiable programming: An end-to-end learning and control framework.NeurIPS, 2020
> [42] Xu, M., et al., Revisiting implicit differentiation for learning problems in optimal control, NeurIPS, 2023

---

> ### Author Response · Authors · 2024-08-09
> **Thank you for your response!**
>
> We sincerely thank the reviewer for reading our rebuttal and for the prompt response.
>
> > One suggestion I would like to make is that DiffTop should be clearly distinguished from the previous works mentioned in the related work section. Currently, the paper’s organization doesn’t make this distinction immediately clear. A dedicated section, perhaps with a simple diagram highlighting the conceptual differences, would be helpful.
>
> Thank you for this suggestion! We agree that dedicated section or a diagram would be especially helpful to clearly distinguish our method from prior works. We will certainly incorporate this in the final version of the paper, in addition to the updated discussion of the closely related works in the initial rebuttal.
>
> We want to thank you again for taking the time to review our paper and read our rebuttal. Your suggestions and feedback have greatly helped improve the quality of our paper.

---

### Official Review · Reviewer_fycy · 2024-07-12

**Soundness:** 3
**Presentation:** 2
**Contribution:** 4
**Rating:** 8
**Confidence:** 4

**Summary:**

The paper introduces DiffTOP, a novel policy class for reinforcement learning (RL) and imitation learning (IL) that utilizes differentiable trajectory optimization to generate policy actions. DiffTOP leverages recent advancements in differentiable trajectory optimization, allowing end-to-end learning of cost and dynamics functions through gradient computation. The approach addresses the "objective mismatch" problem in model-based RL by optimizing dynamics and reward models to directly maximize task performance. For imitation learning, DiffTOP optimizes actions with a learned cost function at test time, outperforming previous methods.

The authors benchmark DiffTOP on 15 model-based RL tasks and 35 imitation learning tasks with high-dimensional inputs like images and point clouds. The results demonstrate that DiffTOP surpasses prior state-of-the-art methods in both domains. The paper includes analysis and ablation studies to provide insights into DiffTOP's learning procedure and performance gains.

**Strengths:**

This paper has several strengths:

- The paper presents a robust and technically rigorous study with extensive experiments and ablation studies across a wide range of challenging environments. The proposed method DiffTOP achieves superior results in both RL and IL tasks with high-dimensional sensory observations.
- The approach effectively tackles the important "objective mismatch" problem inherent in model-based reinforcement learning.
- The ability to compute policy gradients directly with respect to the parameters describing the observation and transition model is a significant advancement, eliminating the need for sample-based estimates.
- DiffTOP alleviates the model mismatch problem by learning the model concurrently with optimization, similar to TD-MPC.

**Weaknesses:**

While the paper has many strengths, there are also some potential weaknesses or areas that could be improved:

- The paper attempts to condense a large amount of information into limited space, which may affect readability and clarity, particularly for readers less familiar with TD-MPC.
- The trajectory optimization solver used (Theseus) does not support constraint optimization, requiring manual unrolling of dynamics instead of presenting it as a constraint to the optimizer.

**Questions:**

- For ManiSkill tasks, have the authors tried more advanced BC baselines (e.g., Diffusion Policy)? It seems the performance of baselines is not very strong.

**Limitations:**

The authors have discussed a few limitations in the paper.

---

> ### Author Rebuttal · Authors · 2024-08-06
>
> We want to extend our heartfelt gratitude for taking the time to review our paper. Thank you for all valuable comments and suggestions on improving the quality of the paper. Below we respond to each of your comments in detail.
>
> **Q: The paper attempts to condense a large amount of information into limited space, which may affect readability and clarity, particularly for readers less familiar with TD-MPC.**
>
> We thank the reviewer for bringing this to our attention. Indeed, the model-based RL part of DiffTOP builds upon TD-MPC, thus requiring some knowledge of TD-MPC for understanding the paper. Following the reviewer’s suggestion, we have updated section 3.2 with more explanations of TD-MPC to add more buildup for introducing our algorithm, and for a better understanding of the paper. We have also updated our paper to include more background information and details on TD-MPC in the appendix for readers less familiar with it.
>
> **Q: The trajectory optimization solver used (Theseus) does not support constraint optimization, requiring manual unrolling of dynamics instead of presenting it as a constraint to the optimizer.**
>
> We thank the reviewer for this feedback. Indeed, Theseus does not support constrained optimization at the time of our paper submission, and thus we have to unroll the dynamics when solving the trajectory optimization problem. The original Theseus paper has the following discussion about constrained optimization in their limitation section: “The nonlinear solvers we currently support apply constraints in a soft manner (i.e., using weighted costs). Hard constraints can be handled with methods like augmented Lagrangian or sequential quadratic programs [99, 100], and differentiating through them are active research topics”. Based on this, it seems that the support of constrained optimization might be added in the future. We leave integrating DiffTOP with constrained optimization as important future work.
>
> **Q: For ManiSkill tasks, have the authors tried more advanced BC baselines (e.g., Diffusion Policy)? It seems the performance of baselines is not very strong.**
>
> We thank the reviewer for this question. ManiSkill tasks all use point cloud as the policy inputs; the original Diffusion Policy paper only tested their method with image inputs, and has not been implemented and tested to work with point cloud inputs. The baseline we compared to in the ManiSkill tasks was the best method that was introduced by the ManiSkill authors, and we have further tuned this baseline method to make the results stronger than presented in their original paper. The improvement of DiffTOP over the ManiSkill baselines in Table 2 provides evidence to the effectiveness of our proposed method.
>
> We also note that we have compared to more advanced BC baselines such as Diffusion Policy and 3D Diffusion Policy (DP3) in other benchmarks where these baseline methods are originally tested, i.e., MetaWorld and RoboMimic, and DiffTOP outperforms them in both benchmarkings,  showing the effectiveness of our proposed method.

---

> > ### Comment · Reviewer_fycy · 2024-08-10
> >
> > Thanks for the detailed response. I don't have further concerns.

---

> ### Author Response · Authors · 2024-08-12
> **Thank you for your response!**
>
> We sincerely thank the reviewer for reading our rebuttal and for the prompt response. We are glad that our rebuttal has addressed your concerns. We want to thank you again for taking the time to review our paper and read our rebuttal. Your suggestions and feedback have greatly helped improve the quality of our paper.

---

### Official Review · Reviewer_kMzm · 2024-07-12

**Soundness:** 3
**Presentation:** 3
**Contribution:** 3
**Rating:** 6
**Confidence:** 4

**Summary:**

The paper presents a method that uses Differentiable Trajectory Optimization as a policy representation. The proposed method extends the work of Temporal difference learning for model predictive control  (TD-MPC) by incorporating a policy-gradient loss for which analytical backpropagation is possible thanks to the differentiable properties of the used trajectory optimization scheme.

The paper evaluates the effectiveness of trajectory optimization as a policy representation for both model-based RL and Imitation Learning, benchmarking the approach in the DeepMind control tasks, the MetaWorld benchmark, the Robomimic benchmark, and the ManiSkill benchmark. Furthermore, the work presents comparisons against other policy representations like traditional feed-forward policies, Energy-based methods (EBMs), and Diffusion Policies.

**Strengths:**

- The paper is well-written, the work is interesting and it presents a simple yet effective extension of TD-MPC.

- The method enables the use of trajectory optimization using high-dimensional observations and it is also able to deal with multimodality in the solution space.

- The method is evaluated thoroughly in different setups (Model-based RL, IL) and against other classes of policy representations, outperforming most baselines in terms of sample efficiency and final reward.

**Weaknesses:**

[Minor]
- The main weakness of the approach is the high computational cost of solving trajectory optimization at test time. The appendix reports 0.052 seconds to infer the action for one timestep (20 Hz), despite the fact that the time horizon is considerably short (1 to 3 steps). Such inference speed might hinder the deployment in certain real-world scenarios.

**Questions:**

- Is there any specific reason why the Levenberg-Marquardt solver was used? Using a second-order solver might lead to fewer iterations to get to a solution, but the time per iteration might also be higher. A simpler solver (SGD) could do the job and be faster, improving the overall training time and inference speed.

- Are there any insights on why increasing the time horizon for the trajectory-optimization policy leads to the slight decrease in performance reported in Appendix A2.3? Typically, a longer time horizon leads to trajectories of better quality. Does the trajectory optimization problem reach convergence for the longer time horizons that were tested?

**Limitations:**

Yes. Limitations of the method are mentioned and I agree with the author's claims that there are no direct potential societal impacts.

---

> ### Author Rebuttal · Authors · 2024-08-06
>
> We want to extend our heartfelt gratitude for taking the time to review our paper. Thank you for all valuable comments and suggestions on improving the quality of the paper. Below we respond to each of your comments in detail.
>
> **Q: [minor] The main weakness of the approach is the high computational cost of solving trajectory optimization at test time. The appendix reports 0.052 seconds to infer the action for one timestep (20 Hz), despite the fact that the time horizon is considerably short (1 to 3 steps). Such inference speed might hinder the deployment in certain real-world scenarios.**
>
> We thank the reviewer for bringing up this issue. We have updated section 5 of the paper to include a more detailed discussion on the inference speed of our method for real-world applications:
>
> “Our current inference speed is 0.052 seconds, which is a control frequency of 20 Hz. We note that such inference speed is comparable or higher to other deep robot learning algorithms that take high-dimensional image or point cloud observations as inputs, e.g., Diffusion Policy [1] reports a control frequency of 10 Hz, and PerAct [2] reports a control frequency of 2.23 Hz. As these methods are shown to be able to be deployed with real-world robots for many tasks, we believe our method’s inference speed of 20 Hz would work well for most real-world robot tasks as well. ”
>
> We would also like to clarify that since we also learn a value function that predicts the future accumulated rewards and use it during the planning process, the policy is able to reason more than 3 steps into the future at inference time.
>
> **Q: Is there any specific reason why the Levenberg-Marquardt solver was used? Using a second-order solver might lead to fewer iterations to get to a solution, but the time per iteration might also be higher. A simpler solver (SGD) could do the job and be faster, improving the overall training time and inference speed.**
>
> We thank the reviewer for this insightful question. We use the Theseus [3] library for differentiating through the trajectory optimization algorithms, and the solvers supported by Theseus include second-order solvers like Gauss-Newton, Levenberg–Marquardt, and Dogleg, and linear solvers such as CHOLMOD. We choose Levenberg-Marquardt as we find it to perform better in early experiments. We didn’t use SGD since it is not supported by the Theseus library, possibly due to its slower convergence rate when dealing with highly non-linear problems. Although implementing SGD to solve the trajectory optimization problem itself is not difficult, robustly differentiating through it may not be trivial and is out of the scope of the current paper. We look forward to trying more solvers, such as SGD, in future work.
>
> **Q: Are there any insights on why increasing the time horizon for the trajectory-optimization policy leads to the slight decrease in performance reported in Appendix A2.3? Typically, a longer time horizon leads to trajectories of better quality. Does the trajectory optimization problem reach convergence for the longer time horizons that were tested?**
>
> We thank the reviewer for this interesting question. We do notice a slight performance drop with longer prediction horizons. The reason could be as follows: since there will always be errors in the learned dynamics function and the reward/value function, there is a tradeoff between the errors in the dynamics model and the errors in the reward/value function when choosing the prediction horizon. With a longer prediction horizon, compounding errors in the learned dynamics model will dominate, whereas with a shorter prediction horizon we expect errors in the learned value function to be more prevalent [4]. The optimal value for this parameter is highly likely to be application-dependent, but as shown in Appendix A2.3, our method can demonstrate robustness to different prediction horizons. When solving the trajectory optimization problem, we run the Levenberg-Marquardt solver for 100 iterations and it has reached convergence within 100 iterations.
>
> [1] Cheng Chi, Siyuan Feng, Yilun Du, Zhenjia Xu, Eric Cousineau, Benjamin Burchfiel, Shuran Song, “Diffusion Policy: Visuomotor Policy Learning via Action Diffusion”, RSS 2023
> [2] Mohit Shridhar, Lucas Manuelli, Dieter Fox, “Perceiver-Actor: A Multi-Task Transformer for Robotic Manipulation”, CoRL 2022
> [3] Luis Pineda, Taosha Fan, Maurizio Monge, Shobha Venkataraman, Paloma Sodhi, Ricky T. Q. Chen, Joseph Ortiz, Daniel DeTone, Austin Wang, Stuart Anderson, Jing Dong, Brandon Amos, Mustafa Mukadam, “Theseus: A Library for Differentiable Nonlinear Optimization”, NeurIPS 2022
> [4] Harshit Sikchi, Wenxuan Zhou, David Held, “Learning Off-Policy with Online Planning”, CoRL 2021

---

### Author Rebuttal · Authors · 2024-08-06

Dear reviewers,

We want to extend our heartfelt gratitude for taking the time to review our paper. Thank you for all valuable comments and suggestions on improving the quality of the paper. We respond to each of your comments in detail in the individual rebuttal.

Following the reviewer's suggestions, the attached PDF contains two additional experiments:
- Comparison to closely related work Amos et al. (Reviewer epML)
- Combining DiffTOP with Dreamer-V3 as an ablation study on the underlying model-based RL algorithm (Reviewer epML)

[1] Amos, B., et al., Differentiable mpc for end-to-end planning and control. NeurIPS, 2018

Best,
Authors

---

### Decision · Program_Chairs · 2024-09-25

**Decision:**

Accept (spotlight)

**Comment:**

The paper presents a method using Differentiable Trajectory Optimization as a policy representation, extending TD-MPC with a policy-gradient loss. It addresses the "objective mismatch" issue by optimizing dynamics models to maximize task performance.

Overall, it's a technically strong paper, demonstrating good performance across diverse tasks. The reviewers had concerns about contextualizing the contribution within existing research, computational efficiency as well as novelty.  However the rebuttal does a good job in clarifying these.

All reviewers lean accept, and the Metra-reviewer finds concurs.
The authors are advised to review the final comments, and update the manuscript accordingly.